# Can Large Language Models Explore In-Context?

**Akshay Krishnamurthy**[1] **Keegan Harris**[2] **Dylan J. Foster**[1] **Cyril Zhang**[1] **Aleksandrs Slivkins**[1]

[1]Microsoft Research      [2]Carnegie Mellon University

keeganh@cs.cmu.edu, {akshaykr,dylanfoster,cyrilzhang,slivkins}@microsoft.com

## Abstract

We investigate the extent to which contemporary Large Language Models (LLMs) can engage in *exploration*, a core capability in reinforcement learning and decision making. We focus on native performance of existing LLMs, without training interventions. We deploy LLMs as agents in simple *multi-armed bandit* environments, specifying the environment description and interaction history entirely *in-context*, i.e., within the LLM prompt. We experiment with GPT-3.5, GPT-4, and LLAMA2, using a variety of prompt designs, and find that the models do not robustly engage in exploration without substantial interventions: i) Only one configuration resulted in satisfactory exploratory behavior: GPT-4 with chain-of-thought reasoning and an externally summarized interaction history; ii) All other configurations did not result in robust exploratory behavior, including those with chain-of-thought reasoning but unsummarized history. While these findings can be interpreted positively, they suggest that external summarization—which may not be possible in more complex settings—is essential for desirable LLM behavior. We conclude that non-trivial algorithmic interventions, such as fine-tuning or dataset curation, may be required to empower LLM-based decision making agents in complex settings.

## 1  Introduction

*In-context learning* is an important emergent capability of Large Language Models (LLMs) whereby one can use a pre-trained LLM to solve a problem by specifying the problem description and relevant data entirely *in-context*, i.e., within the LLM prompt, with no updates to LLM parameters [16]. For example, one can prompt an LLM with numeric covariate vectors and scalar targets and subsequently obtain regression-style predictions from the model by including new covariate vectors in the prompt [28]. Perhaps surprisingly, LLMs are not explicitly trained for this behavior; instead the underlying algorithms employed for in-context learning are extracted from the training corpus and *emerge* at scale.

Since its discovery in the GPT-3 model [16], in-context learning has been actively studied, from theoretical investigations into the underlying mechanisms [e.g., 78, 7] to empirical probes [e.g., 28, 40] to leveraging in-context learning in applications [e.g., 79, 67, 25]. This literature predominantly concerns prediction or supervised learning tasks, and while theoretical progress is in its infancy, our understanding of how to use *in-context supervised learning* (ICSL) in practice is rapidly taking shape.

While ICSL is an important capability, many applications demand the use of ML models for downstream *decision making*. Thus, *in-context reinforcement learning* (ICRL) is a natural next frontier. LLMs are already being used as decision making agents in applications ranging from experimental design in the natural sciences [45] to game playing [63, 72], but our understanding—theoretically and operationally—of ICRL is far less developed than for ICSL. To date, we lack a systematic understanding as to whether LLMs can be considered general-purpose decision-making agents.

Decision making agents must possess three core capabilities: *generalization* (required for supervised learning), *exploration* (making decisions that may be suboptimal in the short term for the sake of gath-

ering more information) and *planning* (to account for long-term consequences of decisions). In this paper, we focus on exploration, the capability to deliberately gather information in order to evaluate alternatives and reduce uncertainty. A recent series of papers [42, 44, 57] demonstrates in-context reinforcement learning behavior (including exploration) in transformer models when they are *explicitly trained* to produce this behavior using data from reinforcement learning agents or expert demonstrations on related tasks. Such training tends to be laborious, expensive, and possibly task-specific. In particular, these findings do not shed light into whether exploratory behavior manifests in general-purpose LLMs obtained via standard training methods, which suggests the following basic question:

*Do contemporary LLMs exhibit the capability to explore in-context?*

**Contributions.** We investigate this question by deploying LLMs as agents in simple synthetic reinforcement learning problems, namely *multi-armed bandits (MABs)* [65, 43], specifying the environment description and interaction history entirely within the LLM prompt. MABs are a well-studied type of RL problem that isolates the tradeoff between exploration and *exploitation*, i.e., making the best decision given the available data. They are also fundamental in that the ability to solve MABs is a prerequisite for more challenging RL tasks. These considerations make MABs a natural choice for systematically studying the in-context exploration abilities of LLMs.

We evaluate the in-context exploration behavior of GPT-3.5 [16], GPT-4 [54], and LLAMA2 [69] in MAB environments, using a variety of prompt designs. In our experiments, we find that only a single configuration (i.e., a prompt design and LLM pair) results in satisfactory exploratory behavior. All other configurations exhibit exploration failures, failing to converge to the best decision (*arm*) with significant probability. We find that this typically happens due to *suffix failures*, where the LLM fails to select the best arm even once after some initial rounds (i.e., in some "time suffix"). This scenario is reflected in Figure 1(a): in particular, GPT-4 with our basic prompt design experiences a suffix failure in $> 60\%$ of the replicates. An alternative failure mode we identify is where the LLM behaves "uniformly", selecting all arms near-equally often and failing to narrow down to the better ones.

The single configuration that succeeds in our experiments involves a combination of GPT-4 and an "enhanced" prompt that (a) provides a suggestive hint to explore, (b) externally summarizes the history of interaction into per-arm averages, and (c) asks the LLM to use zero-shot chain-of-thought reasoning [74, 41]. This configuration is visualized in Figure 1(b). One can interpret this finding positively: state-of-the-art LLMs *do* possess the capability to robustly explore, provided that the prompt is carefully designed to elicit this behavior. On the other hand, the same configuration without external summarization fails, leading to a negative interpretation: LLMs may fail to explore in more complex environments, where external summarization is a non-trivial algorithmic problem.[1]

We conclude that while the current generation of LLMs can perhaps explore in simple RL environments with appropriate prompt engineering, training interventions —in the spirit of Lee et al. [44], Raparthy et al. [57]— may be required to endow LLMs with more sophisticated exploration capabilities required for more complex settings.

**Methodology.** An underlying technical challenge in assessing LLM capabilities and limitations is that one must search a combinatorially large space of prompt designs while obtaining statistically meaningful results, all while meeting the financial and computational constraints associated with LLMs. Assessing in-context bandit learning is even more challenging because (a) stochasticity in the environment demands a high degree of replication for statistical significance and (b) the sample complexity of learning/exploration demands that even a single experiment involve hundreds or thousands of LLM queries to obtain meaningful effect sizes (i.e., separation between successful and failing methods). To address these issues, our core technical contribution is to identify *surrogate statistics* as diagnostics for long-term exploration failure. The surrogate statistics we consider characterize long-term exploration failure, yet can be measured at moderate scale with few replicates and short learning horizons, even when the standard performance measure (namely, reward) is too noisy to be useful.

## 2 Experimental setup

**Multi-armed bandits (MAB).** We consider a basic multi-armed bandit variant, *stochastic Bernoulli bandits*. There are $K$ possible actions (*arms*), indexed as $[K] := \{1, \ldots, K\}$. Each arm $a$ is

---

[1] E.g., if there are many arms, or if we are considering contextual bandits with many contexts, then we may only play each arm (context-arm pair) a few times, so averaging reward separately for each—as we do in our experiments—does not provide much summarization. (See Section 4 for further discussion.)

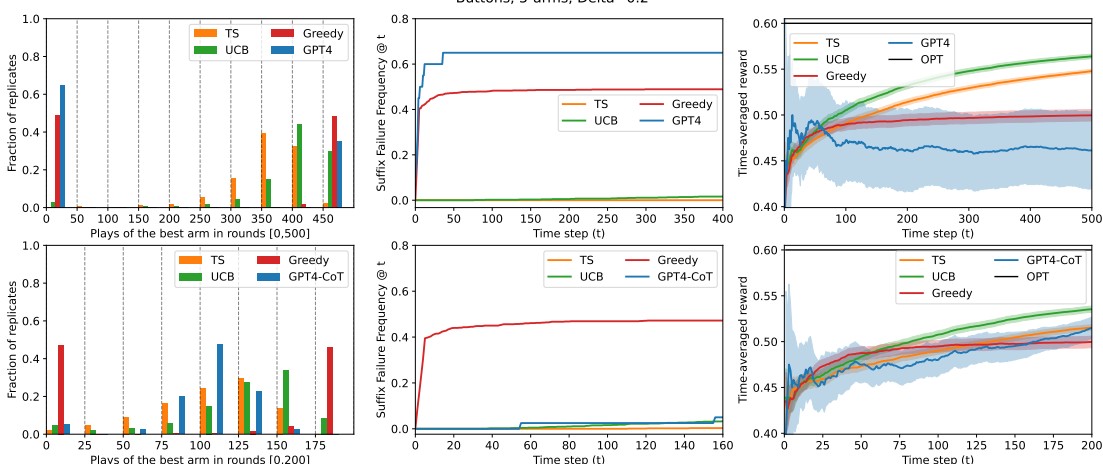

Figure 1: **Representative experiments:** Two prompt configurations for GPT-4 on a 5-armed bandit problem, with exploration failure (top) and success (bottom). The baselines are two standard bandit algorithms with performance guarantees, Upper Confidence Bound (UCB) and Thompson Sampling (TS), as well as the GREEDY algorithm (see Footnote 5). Visualizations are: (Left) histogram over replicates of the number of times the best arm is chosen, (Center) for each $t$, we plot the *suffix failure frequency*, the fraction of replicates for which the best arm is never chosen after time-step $t$, and (Right) cumulative time-averaged rewards, averaged over replicates ($\pm 2$ standard errors).

**(a) Top row.** GPT-4 with our basic prompt design and zero temperature. The experiment runs for $T = 500$ rounds, and is replicated $N = 20$ times, varying environment randomness. We see highly bimodal behavior: a large ($> 60\%$) fraction of replicates pick the best arm only a few times, exhibiting suffix failures similar to GREEDY and very unlike UCB and TS. This is suggestive of a long-term failure to explore; indeed, we see a substantial drop in rewards.

**(b) Bottom row.** GPT-4 with a suggestive framing, summarized history, and chain-of-thought with zero temperature. The experiment runs for $T = 200$ rounds and $N = 40$ replicates. We observe a unimodal distribution of plays of the best arm, very few suffix failures, and reward comparable to TS.

associated with mean reward $\mu_a \in [0, 1]$, which is unknown. An agent interacts with the environment for $T$ time steps, where in each time step $t \in [T]$ the agent selects an arm $a_t \in [K]$ and receives a reward $r_t \in \{0, 1\}$ drawn independently from a Bernoulli distribution with mean $\mu_{a_t}$. Thus, the MAB instance is determined by the mean rewards $(\mu_a : a \in [K])$ and the time horizon $T$. The goal is to maximize the total reward, which roughly corresponds to identifying the *best arm*: an arm with the highest mean reward. A key feature of the MAB setup is that rewards for arms not chosen by the agent are not revealed, so exploration is necessary to identify the best arm.

We focus on MAB instances where the best arm has mean reward $\mu^\star = 0.5 + \Delta/2$ for a parameter $\Delta > 0$, while all other arms have mean reward $\mu = 0.5 - \Delta/2$ (so, $\Delta = \mu^\star - \mu$ is the *gap* between the best and the second-best arm). The main instance we consider has $K = 5$ arms and gap $\Delta = 0.2$. We call this the hard instance, as we also consider an easy instance with $K = 4$ and $\Delta = 0.5$.[2]

**Prompts.** We employ LLMs to operate as decision making agents that interact with MAB instances by prompting them with a description of the MAB problem (including the time horizon $T$) and the history of interaction thus far. Our prompt design allows several independent choices. First is a "scenario", which provides a grounding for the decision making problem, positioning the LLM either a) as an agent choosing *buttons* to press, or b) as a recommendation engine displaying *advertisements* to users. Second, we specify a "framing" as either a) explicitly *suggestive* of the need to balance exploration and exploitation, or b) *neutral*. Third, the history can be presented as a) a *raw* list over rounds, or it can b) be *summarized* via number of plays and average rewards of each arm. Fourth, the requested final answer can be a) a single *arm*, or b) a *distribution* over arms. Finally, we either a) request the answer only, or b) also allow the LLM to provide a "chain-of-thought" (CoT) explanation. Altogether, these choices lead to $2^5 = 32$ prompt designs, illustrated in Figure 2. More details about the prompt design, including examples, are provided in Appendix B.

---

[2]Larger gap $\Delta$ makes it easier to distinguish arms, while smaller $K$ means there are fewer arms to explore.

The most basic prompt design from the options above uses the buttons scenario, neutral framing, and raw history, and requests the LLM to return only an arm with no CoT. Each of the five possible modifications to this prompt can potentially help the LLM, and our experiments evaluate this. For example, both the advertising scenario and suggestive framing might help invoke the LLM's knowledge of bandit algorithms (as bandit algorithms are commonly used in content recommendation). History summarization might help if the LLM cannot reliably summarize history itself (perhaps due to arithmetic errors[3]) and/or does not fully realize that it should. Returning a distribution might help if the LLM can identify a good distribution, but fails to correctly sample from it. Finally, chain-of-thought is known to help in a wide variety of LLM scenarios [74, 50], even when used in a zero-shot manner [41] as we do here.

Prompts are presented to each LLM using both system and user messages (exposed by all three LLM APIs). The system message presents information about the scenario and framing and prompts the LLM about whether to use CoT and whether (and how) to return a distribution. The user message presents the history and reminds the LLM about how to format its response. For GPT-4 only, we found that prompting the LLM to use CoT in the system prompt did not reliably elicit CoT outputs, so—for GPT-4 only—we also consider a *reinforced CoT* prompt design that additionally reminds the LLM to use CoT at the end of the user prompt. See Appendix B for examples.

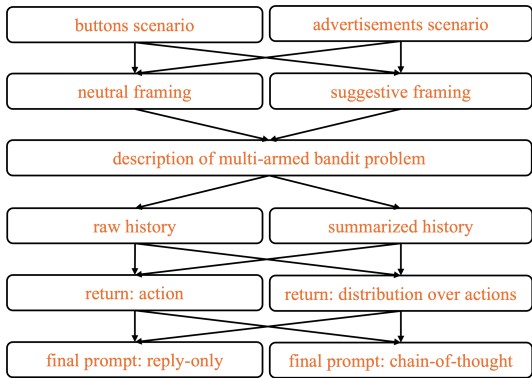

Figure 2: Prompt designs; see Figure 9 for a more detailed view. A prompt is generated by traversing the graph from top to bottom.

**LLM configurations and baselines.** We experiment with three LLMs: GPT-3.5, GPT-4, and LLAMA2.[4] In addition to the prompt variations above, we also consider two choices for the temperature parameter, 0 and 1. A temperature of 0 forces the LLM to be deterministic and therefore isolates the "deliberate" exploration behavior of the LLM itself. A temperature of 1 provides a source of external randomness in the LLM responses, which may or may not result in randomization among the arms. Allowing the LLM to return a distribution instead of a single arm also provides external randomness (as we sample from the returned distribution); to isolate sources of randomness, we do not consider temperature 1 with "return distribution" prompt designs.

We refer to the tuple (prompt design, temperature) as the *LLM configuration*. We identify each configuration with a 5-letter "code" $L_1L_2L_3L_4L_5$, with letters $L_i$ denoting the choices:

- $L_1$: 'B' or 'A' for, resp., buttons or advertisements scenario;
- $L_2$: 'N' or 'S' for, resp., neutral or suggestive framing;
- $L_3$: 'R' or 'S' for, resp., raw or summarized history;
- $L_4$: 'C' or '$\widetilde{\text{C}}$' or 'N' for, resp., chain-of-thought, reinforced CoT, or no CoT.
- $L_5$: '0', '1' or 'D' for, resp., temperature and returning a distribution (with temperature 0).

We refer to "BNRN0" as the *basic* configuration going forward. Most of our experiments consider the "buttons" scenario, and we use the "advertisements" scenario primarily as a robustness check.

For GPT-3.5 and LLAMA2, we do not consider reinforced CoT as it is not required to reliably elicit CoT outputs; thus, we have 48 configurations total. For GPT-4, we primarily used reinforced CoT, but did experiment with some standard CoT prompt designs; thus, there are 72 configurations total.

For baselines, we consider two standard MAB algorithms, UCB [9] and Thompson Sampling (TS) [68], which are optimal in a certain theoretical sense and also reasonably effective in practice. We also consider the GREEDY algorithm, which does not explore and is known to fail.[5] While all three

---

[3]E.g., LLMs sometimes fail at basic arithmetic [27, 48], though this is likely to improve in the near future via better training and/or integrating calculator-like tools.

[4]Specifically: GPT-3.5-TURBO-0613 (released 06/13/2023), GPT-4-0613 (released 06/13/2023), and LLAMA2-13B-CHAT quantized to 4-bits [24].

[5]In each round, GREEDY chooses an arm with the largest average reward so far. It is initialized with one sample of each arm. It *fails* in that with constant probability, it never chooses the best arm after initialization.

baselines have tunable parameters, we perform no parameter tuning (see Section A.1 for a detailed description of each algorithm with parameter settings). In addition to these baselines, some of our experiments include the the $\epsilon$-GREEDY algorithm[6] with various choices of $\epsilon$ to quantitatively demonstrate tradeoffs between exploration and exploitation. We ran 1000 replicates for each baseline and each MAB instance (with rewards realized independently across the replicates).

**Scale of the experiments.** Our main set of experiments has time horizon $T = 100$. To account for randomness in rewards (and possibly in the LLM, via temperature) we ran $N \in \{10, 20\}$ replicates for each LLM configuration and each bandit instance, with rewards generated independently across the replicates. As a robustness check, we ran a single experiment on GPT-4 with the basic configuration for $T = 500$ rounds (with $N = 20$), and obtained consistent/stronger conclusions, see Figure 1(a).

In more detail, for GPT-3.5 we used $N = 20$ replicates across all 48 prompt configurations, resulting in $\approx 200K$ queries in total. GPT-4 was an order of magnitude more expensive, considerably slower on throughput, and subject to unpredictable throttling. As such, we only used $N = 10$ replicates across 10 representative prompt configurations.[7] For additional robustness checks, we ran four GPT-4 configurations with $T = 200$, two for $N = 20$ replicates and two for $N = 40$ replicates. In total, this resulted in $\approx 50K$ queries issued to GPT-4. LLAMA2 was essentially free from our perspective (since it was locally hosted), but its performance was consistently sub-par; we limited our experiments to the hard MAB instance, 32 configurations, and $N = 10$ replicates.

We emphasize that bandit experiments with LLMs are quite costly in terms of money and time. They take $N \cdot T$ LLM queries for each LLM configuration and each MAB instance being tested. Both $N$ and $T$ must be relatively large to obtain statistically meaningful results: $N$ governs the significance level and must be large to overcome randomness in reward realizations, while $T$ governs the effect size and must be large so that good algorithms have enough time to identify the optimal arm. Both issues are more pronounced in harder MAB instances (many arms $K$ and/or small gap $\Delta$), but exploration failures also tend to be less frequent in (very) easy MAB instances. Further, we need to cover the space of possible prompt designs, which is essentially infinitely large, to ensure that our findings do not overfit to one particular design. Thus, ideally we would take $N, T$, the number of MAB instances, and the number of prompts to be rather large, but doing so is not practically feasible.[8] Instead, we use moderately small gap $\Delta = 0.2$, moderately large choices for $N \in \{10, 20\}$ and $T = 100$, and the prompt design space as described above.

As we see below, these choices ($N \in \{10, 20\}$, $T = 100$, $\Delta = 0.2$) do not provide enough statistical power to distinguish between successful and unsuccessful methods based solely on accumulated rewards. In lieu of further increasing the scale of the experiments, which is not practically feasible, we rely on *surrogate statistics* which can be detected at our moderate scale, and are highly suggestive of long-term/persistent exploration failures. Our robustness checks with larger $T$ and $N$, as well as qualitative findings that we report below provide supporting evidence for this methodology.

## 3 Experimental results

In this section, we present our experimental findings, beginning with a summary. In Section 3.1 we investigate failing LLM configurations in detail. In Section 3.2, we focus on the single successful LLM configuration we identified. In Section 3.3, we attempt to diagnose root causes for failures.

**Overview.** All but one LLM configurations considered exhibit exploration failures, not converging to the best arm with significant probability. This happens either due to *suffix failures*, where the LLM never selects the best arm after a small number of initial rounds, or (in a few configurations) due to *uniform-like failures*, where the LLM selects all arms at an approximately uniform rate, failing to eliminate poorly performing arms. The one exception is GPT-4 with the $\text{BSS}\widetilde{\text{C}}0$ configuration, i.e., the buttons scenario, suggestive framing, summarized history, reinforced CoT, and temperature $0$.

We summarize our key findings in Figures 3-4. Figure 3 summarizes the main set of experiments (on the hard MAB instance), mapping each LLM configuration to a single point on a scatter plot.

---

[6]$\epsilon$-GREEDY is a standard MAB algorithm which in each round chooses an arm uniformly at random with a given probability $\epsilon$, and exploits (i.e., mimics GREEDY) otherwise.

[7]$N = 10$ for the buttons scenario, and $N = 3$ for the robustness check with the advertisements scenario.

[8]Raw-history prompts and chain-of-thought outputs are particularly expensive, as LLM APIs bill per token.

| | TS | UCB | Greedy | BNRN0 | BNRN1 | BNRND | BNRC0 | BNSN0 | BSRN0 | BSSC0 | BSSC1 | BSSCD | BSSC̃0 |
|---|---|---|---|---|---|---|---|---|---|---|---|---|---|
| MedianReward | 0.47 | 0.55 | 0.40 | 0.63 | 0.70 | 0.33 | 0.35 | 0.60 | 0.45 | 0.68 | 0.28 | 0.37 | 0.47 |
| SuffFailFreq(T/2) | 0.01 | 0.02 | 0.48 | 0.50 | 0.40 | 0.00 | 0.50 | 0.60 | 0.70 | 0.30 | 0.20 | 0.00 | 0.00 |
| K*MinFrac | 0.28 | 0.18 | 0.05 | 0.03 | 0.04 | 0.41 | 0.09 | 0.07 | 0.05 | 0.09 | 0.19 | 0.49 | 0.33 |
| GreedyFrac | 0.62 | 0.76 | 1.00 | 0.52 | 0.46 | 0.45 | 0.78 | 0.99 | 0.59 | 0.93 | 0.88 | 0.49 | 0.69 |

Figure 4: GPT-4 for $T = 100$: a per-configuration **summary table** on the hard MAB instance with $N = 10$ replicates. Only three GPT-4 configurations do not exhibit suffix failures; two of these (BNRND and BSSCD) exhibit uniform-like failures. The final configuration (BSSC̃0) succeeds.

The axes correspond to two *surrogate statistics*, SuffFailFreq and $K \cdot$ MinFrac, which represent the strength of the two failure modes (suffix failures and uniform-like failures), and are described in detail in the sequel. Figure 4 displays SuffFailFreq, MinFrac, GreedyFrac (which measures how similar a method is to GREEDY), and additional summary statistics for each GPT-4 configuration in the main set of experiments. These statistics reveal that all of the LLM configurations, except for GPT-4-BSSC̃0 (the blue star in Figure 3), behave fundamentally differently from the baseline algorithms UCB and TS, and we find that these differences result in a large, persistent drop in performance. Conversely, we find that GPT-4-BSSC̃0 successfully explores and (hence) converges to the best arm.

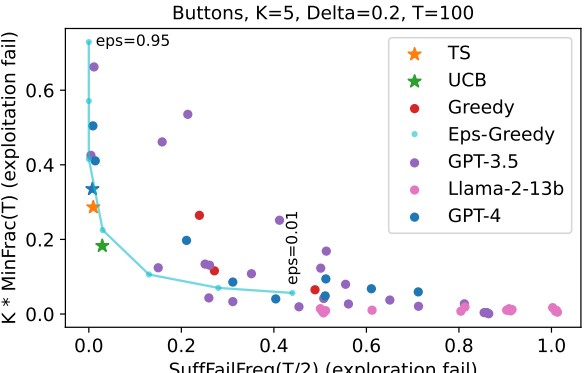

Figure 3: Scatter plot summarizing all experiments with $T = 100$. We plot suffix failures (via SuffFailFreq($T/2$)) vs. uniform-like failures (via $K \cdot$ MinFrac($T$)). Each LLM/configuration pair maps to a dot (some dots overlap). The only successful GPT-4 configuration (BSSC̃0) is labeled with a star. We also plot $\epsilon$-GREEDY, tracing out the tradeoffs for different $\epsilon$.

## 3.1 Identifying failures

We now give a precise overview of the exploration failures illustrated in Figure 3 and Figure 4, and provide additional results and figures that illustrate failure in greater detail. We focus on GPT-4, as GPT-3.5 and LLAMA2 perform worse (and often *much* worse) in all experiments; detailed results for GPT-3.5 and LLAMA2 are included in Appendix C. We begin with detailed background on the surrogate statistics, SuffFailFreq and MinFrac, used to quantify failures in Figures 3 and 4 and beyond, providing evidence that exploration failure—as quantified by these statistics—results in a persistent drop in performance.

**Suffix failures.** Most of the LLM configurations we consider exhibit highly *bimodal* behavior, whereby a large fraction of the replicates choose the best arm very rarely, and a few replicates converge to the best arm extremely quickly. Consistent with this bimodal behavior, we observe a large incidence of *suffix failures*, where the best arm is not selected even once after a small number initial of rounds (i.e., in some "time suffix"). Suffix failures are suggestive of a long-term failure to explore which cannot be improved by running the algorithm for longer, because, without playing the optimal arm, one cannot acquire information to learn that it is indeed optimal. Such behaviors are qualitatively similar to those of GREEDY and qualitatively very different from those of UCB and Thompson Sampling.

Our surrogate statistic for measuring suffix failures is defined as follows: For an experiment replicate $R$ and round $t$, let SuffFail($t, R$) be a binary variable that is 1 if the best arm is never chosen in rounds $[t, T]$. Then let SuffFailFreq($t$) := mean({SuffFail($t, R$) : replicates $R$}). Suffix failures manifest in most of our experiments at $T = 100$. In the scatter plot in Figure 3, the X-axis plots SuffFailFreq($T/2$) for each LLM configuration, and we find that all but five configurations

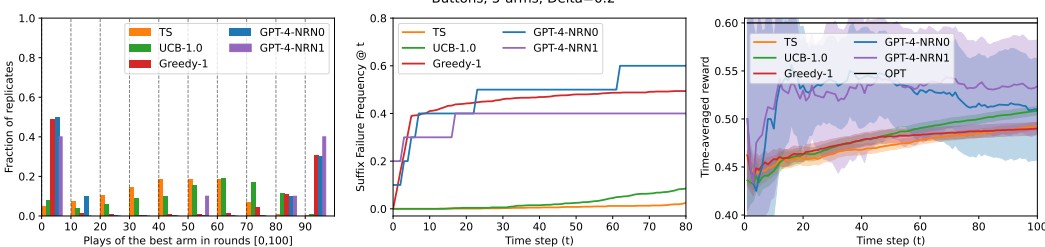

Figure 5: Bimodal behavior and suffix failures for GPT-4 with $T = 100$, same visualizations as in Figure 1. Shown: the basic configuration (BNRN0) and the ablation with temperature 1 (BNRN1).

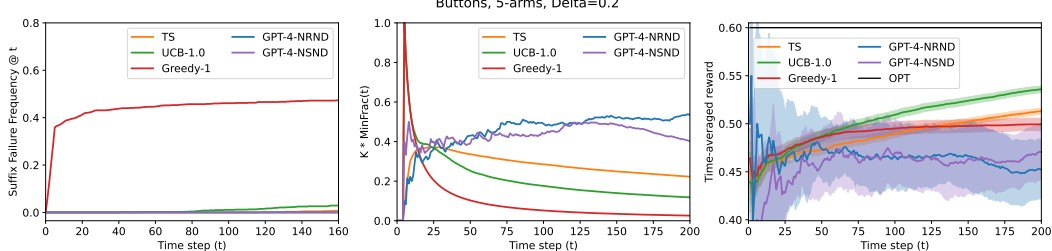

Figure 6: Detailed view of uniform-like failures for GPT-4 (the BNRND and BNSND configurations) with $T = 200$. Visualizations are: (Left) suffix failure frequency, (Center) $K \cdot \texttt{MinFrac}(t)$ as a function of $t$ and (Right) cumulative time-averaged rewards. These configurations exhibit uniform-like failures but not suffix failures, and uniform-like failures are detrimental to long-term rewards.

have $\texttt{SuffFailFreq}(T/2) \geq 15\%$. Recalling the definition of suffix failures, this means that $\geq 15\%$ of the time, these configurations do not pull the best arm *even once* in the last half of the rounds.

A more detailed view of suffix failures and bimodal behavior can be obtained by focusing on individual LLM configurations. We visualize this for the basic configuration (GPT-4-BNRN0) in Figure 1 (top) for $T = 500$, and in Figure 5 for GPT-4 (BNRN0 and BNRN1) at $T = 100$. In these detailed views, the middle panels plot $\texttt{SuffFailFreq}(t)$ at each time $t$ for the given LLM configurations, as well as UCB, TS, and GREEDY. We find that these LLM configurations have much higher suffix failure rates than both UCB and TS. Bimodal behavior is visualized in the left panel of each plot, where for each configuration, a large fraction of replicates rarely pulls the best arm, while the remaining fraction almost always pulls the best arm. Because of this bimodal behavior (particularly because a constant fraction of replicates by chance almost always pull the best arm), suffix failures are not fully reflected in the total reward plots in the right panels of Figure 5, since the time horizon $T = 100$ is not large enough. However, as mentioned, suffix failures are suggestive of an irrecoverable failure to explore which leads to stark differences in reward for larger $T$. This is precisely what we find at $T = 500$ in Figure 1, which suggests that suffix failures indeed lead to poor long-term performance.

**Uniform-like failures.** Returning to the left panel of Figure 3, we see that three GPT-4 configurations avoid suffix failures. Two of these configurations exhibit a different type of failure, where the LLM selects arms in roughly equal proportions for the entirety of the $T$ rounds and fails to exploit the acquired information to focus on the better arms. We call this a *uniform-like failure*.

Our surrogate statistic for measuring such failures is defined as follows: For a particular experiment replicate $R$ and round $t$, let $f_a(t, R)$ be the fraction of rounds in $[1, t]$ in which a given arm $a$ is chosen, $\texttt{MinFrac}(t, R) := \min_a f_a(t, R)$, and $\texttt{MinFrac}(t) := \text{mean}(\{\texttt{MinFrac}(t, R) : \text{replicates } R\})$. Since $\texttt{MinFrac}(t) \leq 1/K$, $\forall t \in [T]$, we always plot $K \cdot \texttt{MinFrac}(t)$, so as to rescale the range to $[0, 1]$. Larger $\texttt{MinFrac}(t)$ corresponds to a more uniform selection of arms at time $t$. When an LLM's $\texttt{MinFrac}(t)$ does not decrease over time and stays substantively larger than that of the baselines (especially as $t$ approaches the time horizon $T$), we take it as an indication of a uniform-like failure.

The Y-axis of Figure 3 records $K \cdot \texttt{MinFrac}(T)$ for each configuration, where we see that of the three GPT-4 configurations that avoid suffix failures, two configurations have very high $\texttt{MinFrac}(T)$ relative to UCB and TS (the third configuration is GPT-4-BSS$\widetilde{\text{C}}$0, which is successful). These two

configurations are GPT-4-BNRND and GPT-4-BSSCD, both of which use the *distributional* output format. We provide a more detailed view of GPT-4-BNRND (as well as GPT-4-BNSND, which also exhibits uniform-like failures, but only differs from GPT-4-BNRND in the use of summarized history) in Figure 6, which considers a longer horizon and more replicates ($T = 200$ and $N = 20$). The middle panel reveals that $K \cdot \text{MinFrac}(t)$ does not decrease over time for these LLM configurations, while it does for the baselines. This behavior results in no suffix failures, but leads to much lower reward than the baselines. In particular, we obtain a clear separation in total reward, showing that uniform-like failures indeed result in poor long-term performance.

**Generality of the failures.** To summarize, Figure 3 shows that all LLM configurations except GPT-4-BSS$\widetilde{\text{C}}$0 exhibit either a suffix failure or a uniform failure for the hard MAB instance and the buttons scenario. Scatter plots for the other three experiments (i.e., the advertisements scenario and/or the easy MAB instance) are qualitatively similar and are deferred to Appendix C.

The same data, but with attributions to specific LLM configurations, are presented for *all* GPT-4 configurations in Figure 4; analogous tables for other LLMs and experimental settings are given in Appendix C. As it is not instructive to present detailed plots such as Figure 5 for every LLM configuration, Figure 4 summarizes the performance of each configuration with just a few statistics. We include: $\text{SuffFailFreq}(T/2)$ and $\text{MinFrac}(T)$, defined above; $\text{MedianReward}$: the rescaled median (over replicates) of the time-averaged total reward;[9] $\text{GreedyFrac}$: the fraction of *greedy rounds* (where an arm with a largest average reward is selected), averaged over the replicates. $\text{GreedyFrac}$ is one way to quantify the extent to which a configuration behaves like GREEDY.

We now summarize further findings from the scatter plots (Figures 3 and 10) and the summary tables (Figures 11 to 17). First, GPT-4 performs much better than GPT-3.5, and LLAMA2 performs much worse (in particular, the suffix failure frequency for LLAMA2 ranges from that of GREEDY to much larger). Second, we observe that all LLMs are sensitive to small changes in the prompt design. However, the different modifications we consider appear to interact with each other, and it is difficult to identify which individual modifications improve performance and which degrade it.

### 3.2 Investigating successes

On the hard MAB instance, the only configuration in our experiments that avoids both suffix failures and uniform-like failures is GPT-4 with the BSS$\widetilde{\text{C}}$0 prompt design. As can be seen from Figure 4, at $T = 100$, this configuration has no suffix failures, the $K \cdot \text{MinFrac}$ value is only slightly larger than TS, and the reward is comparable to TS. These statistics suggest that this configuration succeeds.

For more statistically meaningful results supporting this claim, we run GPT-4-BSS$\widetilde{\text{C}}$0 on the hard MAB instance with $T = 200$ and $N = 40$. We also consider GPT-4-BSR$\widetilde{\text{C}}$0, which swaps summarized history for raw history, as an ablation. Figure 7 summarizes this experiment, while Figure 1(b) provides a detailed view of the BSS$\widetilde{\text{C}}$0 configuration.

| | TS | UCB | Greedy | BSR$\widetilde{\text{C}}$0 | BSS$\widetilde{\text{C}}$0 |
|---|---|---|---|---|---|
| MedianReward | 0.59 | 0.70 | 0.60 | 0.65 | 0.54 |
| SuffFailFreq(T/2) | 0.00 | 0.02 | 0.47 | 0.12 | 0.03 |
| K*MinFrac | 0.23 | 0.12 | 0.03 | 0.11 | 0.29 |
| GreedyFrac | 0.66 | 0.81 | 1.00 | 0.75 | 0.68 |

Figure 7: Summary statistics of two GPT-4 configurations with reinforced CoT (BSR$\widetilde{\text{C}}$0 and BSS$\widetilde{\text{C}}$0), on the hard MAB instance with $T = 200$ and $N = 40$ replicates. BSR$\widetilde{\text{C}}$0 shows suffix failures. BSS$\widetilde{\text{C}}$0 has neither suffix nor uniform-like failures and reasonable reward.

We see that BSS$\widetilde{\text{C}}$0 continues to avoid suffix failures and perform relatively well in terms of reward for larger $T$. On the other hand, the ablation BSR$\widetilde{\text{C}}$0 exhibits a non-trivial fraction of suffix failures, a fundamentally different behavior.

We provide additional visualizations with some qualitative evidence toward the success of BSS$\widetilde{\text{C}}$0, as well as the failure of other configurations. In Figure 8, we plot the fraction of rounds in $[0, t]$ where the optimal arm was pulled; we plot this for individual replicates, as a function of $t$. BSR$\widetilde{\text{C}}$0 is

---

[9]Specifically, let $\Phi(R)$ be the time-averaged total reward for a given replicate $R$. Then $\mathbb{E}[\Phi(R)]$ ranges over $[1/2 - \Delta/2, \ 1/2 + \Delta/2]$. We rescale $\Phi(R)$, by translating and multiplying, so that $\mathbb{E}[\Phi(R)]$ ranges in $[0, 1]$.

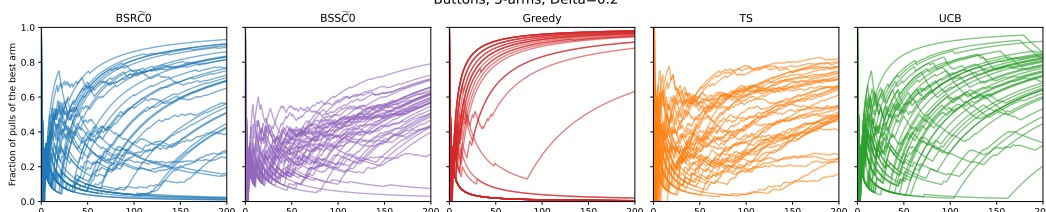

Figure 8: Per-replicate behavior: two reinforced-CoT GPT-4 configurations & the baselines. For each algorithm, replicate and round $t$, we plot the fraction of rounds in $[0, t]$ when the best arm was pulled.

visually similar to UCB, except that a non-trivial fraction of runs exhibit suffix failures (the curves that converge to 0 on the plot). Meanwhile, BSS$\widetilde{C}$0 is visually similar to TS, with almost all replicates slowly converging to 1. Another visualization, presented in Appendix D, shows the arm chosen at each time step for particular replicates; this is for several LLM configurations ("successful" and not), as well as the baselines. These visualizations, along with the summary statistics, suggest that BSS$\widetilde{C}$0 behaves most similarly to TS, which further suggests a similar convergence in the long run.

### 3.3 Root causes

*Why* do LLMs behave the way they do? Particularly, can one explain their failures via flaws in their *per-round* decisions? Two natural hypotheses are that the failing LLM configurations are either a) too greedy, or b) too uniform-like. Indeed, most GPT-4 configurations behave much like GREEDY on the easy MAB instance; yet, they avoid suffix failures and accrue large rewards, and so does GREEDY. However, on the hard instance, most GPT-4 configurations seem to be doing something non-trivial.

A secondary experiment studies this further: Each agent (LLM or baseline) faces a "data source" (distribution of bandit histories) and makes a single decision. We used GPT-3.5 and several data sources. We find it difficult to separate LLMs from the baselines based on the per-round performance, as the latter is very sensitive to the data source. While a deeper investigation is needed, we report this difficulty as a non-trivial finding. All these results are discussed in Appendix E.

## 4 Discussion and open questions

Our investigation suggests that contemporary LLMs do not robustly engage in exploration required for very basic statistical RL and decision making problems, at least without further intervention. Let us identify several natural next steps. First, *experiment with other prompts:* as in many other settings [61], small changes to our prompt template might improve performance; but sensitivity to prompt design is already concerning. Second, *experiment with few-shot prompting,* where the prompt contains examples of exploratory behavior, or use such examples to *fine-tune* the LLM. Third, *train the LLM to use auxiliary tools,* such as a calculator for basic arithmetic or a "randomizer" to correctly sample from a distribution. We emphasize that cost, access to models, and compute pose significant barriers to further study, particularly because of the need to employ long horizons $T$ and many replicates $N$ to obtain statistically meaningful results. To this end, we believe that further methodological and/or statistical advancements to enable cost-effective diagnosis and understanding of LLM-agent behavior (e.g., our surrogate statistics) are essential.

**Implications for more complex problems.** Our focus on simple MAB problems provides a clean and controllable experimental setup to study the exploratory behavior of LLMs. Exploration failures here suggest that similar failures will also occur in more complex RL and decision-making settings. On the other hand, mitigations must be developed with caution, as solutions that succeed for the MAB setting may not generalize to more complex settings. For example, while GPT-4 with summarized interaction history and reinforced CoT seems to successfully explore in our MAB setting, it is not clear how one should externally summarize the history in settings with complex, high-dimensional observations such as contextual bandits (see Footnote 1). Indeed, even for linear contextual bandits, the approach may not be applicable without a substantial algorithmic intervention (such as, e.g., a linear regression computed externally and included in the prompt) and the many explicit modeling and algorithmic choices involved therein. We believe a deeper investigation of algorithmic interventions is essential to understand the extent to which LLMs can operate as decision-making agents.

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

# A  Related work

This paper belongs to a recent body of work that aims to understand the capabilities of LLMs, i.e., what they can and cannot do well, and why. Capabilities that have received considerable attention, but are peripheral to the present paper, include general intelligence [18, 14], causal [39, 81] and mathematical reasoning [21, 49], planning [70, 51, 15], and compositionality [82].

In more detail, our work contributes to the broader literature on capabilities of in-context learning. Prior studies of in-context learning include theoretical [78, 7, 84, 1, 83, 31, 20, 4, 75, 26, 76, 36, 34, 46, 71, 11, 30, 37] and empirical [28, 40, 6, 32, 58, 73, 13, 29, 62, 8] investigations, though as mentioned in the prequel, the vast majority of this work pertains to in-context supervised learning; in-context reinforcement learning has received far less attention. The small collection of empirical works that study in-context RL [42, 44, 57, 79] focus on models trained from scratch using trajectory data collected from another agent (either an RL algorithm or an expert); theoretically, Lee et al. [44] and later Lin et al. [47] justify this approach with a Bayesian meta-reinforcement learning perspective [64], and show that pre-trained transformers can implement classical exploration strategies like Thompson sampling and upper confidence bounds (UCB). However, these works require interventions to the *pre-training* phase of the language model, and do not study whether existing LLMs exhibit exploration capabilities under standard training conditions.

Perhaps closest to the present paper, Coda-Forno et al. [22] evaluates the performance of in-context learning with GPT-3.5 on a two-armed bandit task and an associated meta-learning task. As with our study, they find that GPT-3.5 performs similarly (in fact, slightly worse) than GREEDY; however, they do not consider long enough time horizons to distinguish GREEDY from successful baselines like UCB.

In parallel, there is a rapidly growing line of work that applies LLMs to real-world decision-making applications. Beyond previously mentioned works [63, 72, 45], which consider applications to gaming, programming, and medicine, highlights include Park et al. [56], who introduce generative agents which simulate human behavior in an open-world environment, Ahn et al. [5], Xu et al. [80], who develop LLM-enabled robots.

**Concurrent work.** Two closely related concurrent works [77, 55] also study in-context LLM performance in bandit tasks. Wu et al. [77] considers a battery of tasks that aim to characterize "intelligent agents" with two-armed bandits as a specific task of interest. Their bandit experiments differ in several key respects: They consider a very easy MAB instance (with 2 arms and a gap $\Delta = 0.6$, which is much easier than both of our instances), focus on a single prompt design (similar to our basic prompt), and compare to human players rather than algorithmic benchmarks. These differences lead to very different experimental findings. In particular, they find that GPT-4 performs well on their simple MAB instance, converging very quickly to the best arm, while we find that GPT-4 with a similar prompt fails on a harder MAB instance. However, their finding is consistent with ours, as we also find that several configurations of GPT-4 do well on the easy MAB instance. As we discuss in Section 3.3, this instance is too simple to provide compelling evidence for principled exploratory behavior.

Park et al. [55] primarily focus on full-information online learning and repeated game settings but also evaluate LLMs in bandit settings. Their experiments differ from ours in two significant ways. First, although some of their data generation protocols are stochastic in nature, they are primarily interested in adversarial settings. Consequently they compare with adversarial bandits baselines and present the history to the LLM via importance-weighted losses [10]. Second, they mostly consider shorter time horizons ($T = 25$ for bandits and up to $T = 50$ for full-information). In an updated version of their paper (announced on arXiv in Fall 2024), they also include longer horizon experiments of their original settings, where they find that LLMs continue to perform well, as well as experiments with our hard MAB instance with horizon $T = 100$, where they evaluate uniform and suffix failures. Interestingly, they find that both GPT-4 and GPT-4O succeed (with high reward, no suffix failures, and low MinFrac) when using their default prompt which asks for distributional output, chain-of-thought, and which presents the history via importance weighting. They further find that removing importance weighting results in failures, specifically, higher MinFrac for GPT-4 and suffix failures for GPT-4O. These findings are perhaps consistent with ours: both results highlight that pre-processing the history (either via summarization or via importance weighting) is crucial for eliciting exploratory behavior from LLMs.

Other concurrent work includes Schubert et al. [60], Hayes et al. [33], Coda-Forno et al. [23] who use in-context bandit and other tasks to study whether LLMs exhibit human-like behavior (particularly, biases) in decision making tasks.

We also refer the interested reader to a recent survey of methods for using LLMs in reinforcement learning settings [19].

**Follow-up work.** Monea et al. [52] and Nie et al. [53] follow up on our results with several new experimental findings. Both works consider *contextual* bandits (and Nie et al. [53] also considers vanilla MAB), and find that LLMs fail to explore without non-trivial interventions. In this sense, these works corroborate our main findings. Further, both works propose interventions that improve LLM exploration. In particular, Monea et al. [52] propose a training-free intervention whereby the interaction history is subsampled uniformly before it is included in the LLM prompt, while Nie et al. [53] consider few-shot prompting and fine-tuning with optimal demonstrations. These interventions improve performance, but are still not competitive with standard algorithmic baselines.

## A.1   Further background on multi-armed bandits

Here, we provide additional background on the multi-armed bandit problem, and on the baseline algorithms used in this paper. Deeper discussion can be found in Bubeck and Cesa-Bianchi [17], Slivkins [65], Lattimore and Szepesvári [43].

The UCB algorithm [9] explores by assigning each arm $a$ an *index*, defined as the average reward from the arm so far plus a *bonus* of the form $\sqrt{C/n_a}$, where $C = \Theta(\log T)$ and $n_a$ is the number of samples from the arm so far. In each round, it chooses an arm with the largest index. The bonus implements the principle of *optimism under uncertainty*. We use a version of UCB that sets $C = 1$ (a heuristic), which has been observed to have a favorable empirical performance [e.g., 66, 35].

Thompson Sampling [68, 59, for a survey] proceeds as if the arms' mean rewards were initially drawn from some Bayesian prior. In each round, it computes a Bayesian posterior given the history so far, draws a sample from the posterior, and chooses an arm with largest mean reward according to this sample (i.e., assuming the sample were the ground truth). In our setting, the prior is essentially a parameter to the algorithm. We choose the prior that draws the mean reward of each arm independently and uniformly at random from the $[0, 1]$ interval. This is one standard choice, achieving near-optimal regret bounds, as well as good empirical performance [38, 2, 3]. Each arm is updated independently as a Beta-Bernoulli conjugate prior. Further optimizing UCB and Thompson Sampling is non-essential to this paper, as they already perform quite well in our experiments.

Provable guarantees for bandit algorithms are commonly expressed via *regret*: the difference in expected total reward of the best arm and the algorithm. Both baselines achieve regret $O(\sqrt{KT \log T})$, which is nearly minimax optimal as a function of $T$ and $K$. They also achieve a nearly instance-optimal regret rate, which scales as $O\left(K/\Delta \, \log T\right)$ for the instances we consider.

The $\epsilon$-GREEDY algorithm (Footnote 6) is fundamentally inefficient in that it does not adaptively steer its exploration toward better-performing arms. Accordingly, its regret rate scales as $T^{2/3}$ (for an optimal setting of $\epsilon \sim T^{-1/3}$). Fixing such $\epsilon$, regret does not improve for easier instances.

The GREEDY algorithm (Footnote 5) does not explore at all, which causes suffix failures. This is obvious when the algorithm is initialized with a single sample ($n = 1$) of each arm: a suffix failure happens when the good arm returns 0, and one of the other arms returns 1. However, suffix failures are not an artifact of small $n$: they can happen for any $n$, with probability that scales as $\Omega(1/\sqrt{n})$ [12].

# B  Prompt designs

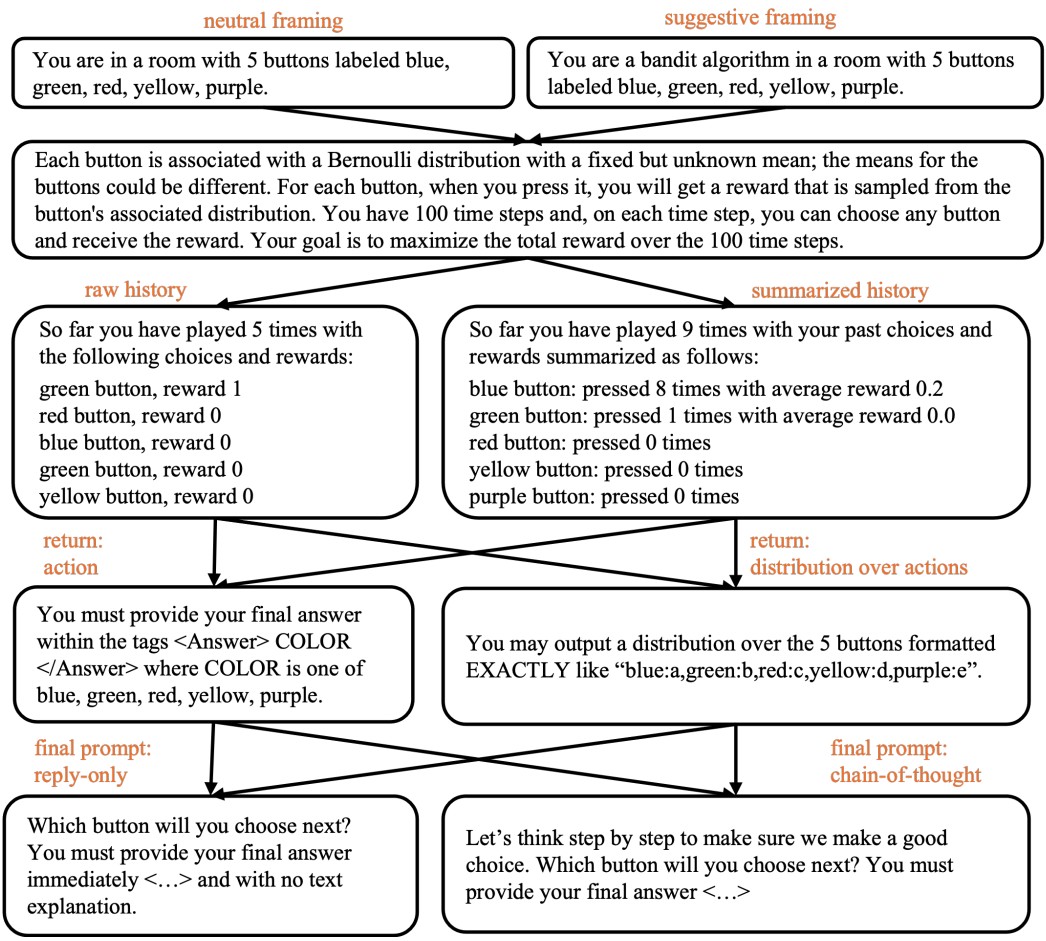

**neutral framing**

You are in a room with 5 buttons labeled blue, green, red, yellow, purple.

**suggestive framing**

You are a bandit algorithm in a room with 5 buttons labeled blue, green, red, yellow, purple.

Each button is associated with a Bernoulli distribution with a fixed but unknown mean; the means for the buttons could be different. For each button, when you press it, you will get a reward that is sampled from the button's associated distribution. You have 100 time steps and, on each time step, you can choose any button and receive the reward. Your goal is to maximize the total reward over the 100 time steps.

**raw history**

So far you have played 5 times with the following choices and rewards:

green button, reward 1
red button, reward 0
blue button, reward 0
green button, reward 0
yellow button, reward 0

**summarized history**

So far you have played 9 times with your past choices and rewards summarized as follows:

blue button: pressed 8 times with average reward 0.2
green button: pressed 1 times with average reward 0.0
red button: pressed 0 times
yellow button: pressed 0 times
purple button: pressed 0 times

**return: action**

You must provide your final answer within the tags <Answer> COLOR </Answer> where COLOR is one of blue, green, red, yellow, purple.

**return: distribution over actions**

You may output a distribution over the 5 buttons formatted EXACTLY like "blue:a,green:b,red:c,yellow:d,purple:e".

**final prompt: reply-only**

Which button will you choose next? You must provide your final answer immediately <…> and with no text explanation.

**final prompt: chain-of-thought**

Let's think step by step to make sure we make a good choice. Which button will you choose next? You must provide your final answer <…>

Figure 9: Prompt designs with text in the buttons scenario, expanding on Figure 2.

### B.1 Prompt examples

Let us present three full examples of our prompts. We remove the blank lines for the sake of readability.

**(a)** Our basic prompt design (i.e., BNRN, as temperature is specified elsewhere): the buttons scenario with neutral framing and raw histories, asking the LLM to return an action without chain-of-thought reasoning.

> [SYSTEM] You are in a room with 5 buttons labeled blue, green, red, yellow, purple. Each button is associated with a Bernoulli distribution with a fixed but unknown mean; the means for the buttons could be different. For each button, when you press it, you will get a reward that is sampled from the button's associated distribution. You have 10 time steps and, on each time step, you can choose any button and receive the reward. Your goal is to maximize the total reward over the 10 time steps.
>
> At each time step, I will show you your past choices and rewards. Then you must make the next choice, which must be exactly one of blue, green, red, yellow, purple. You must provide your final answer immediately within the tags <Answer>COLOR</Answer> where COLOR is one of blue, green, red, yellow, purple and with no text explanation.
>
> [USER] So far you have played 2 times with the following choices and rewards:
>
> blue button, reward 1
> green button, reward 0
>
> Which button will you choose next? Remember, YOU MUST provide your final answer within the tags <Answer>COLOR</Answer> where COLOR is one of blue, green, red, yellow, purple.

**(b)** The adverts scenario with all the "advanced" modifications (i.e., ASSCD) : suggestive framing, summarized histories, asking the LLM to return a distribution over actions, and asking for a chain-of-thought reasoning.

> [SYSTEM] You are recommendation engine that chooses advertisements to display to users when they visit your webpage. There are 5 advertisements you can choose from, named A, B, C, D, E. When a user visits the webpage you can choose an advertisement to display and you will observe whether the user clicks on the ad or not. You model this by assuming that each advertisement has a certain click rate and users click on advertisements with their corresponding rates.
>
> You have a budget of 10 users to interact with and your goal is to maximize the total number of clicks during this process.
>
> A good strategy to optimize for clicks in these situations requires balancing exploration and exploitation. You need to explore to try out all of the options and find those with high click rates, but you also have to exploit the information that you have to accumulate clicks.
>
> When each user visits the webpage, I will show you a summary of the data you have collected so far.
>
> Then you must choose which advertisement to display. You may output a distribution over the 5 choices formatted EXACTLY like "A:n1,B:n2,C:n3,D:n4,E:n5".
>
> Let's think step by step to make sure we make a good choice. Then, you must provide your final answer within the tags <Answer>DIST</Answer> where DIST is the distribution in the format specified above.
>
> [USER] So far you have interacted with 2 users. Here is a summary of the data you have collected:
>
> Advertisement A was shown to 1 users with an estimated click rate of 1.00
> Advertisement B was shown to 1 users with an estimated click rate of 0.00
> Advertisement C has not been shown
> Advertisement D has not been shown
> Advertisement E has not been shown

Which advertisement will you choose next? Remember, YOU MUST provide your final answer within the tags <Answer>DIST</Answer> where DIST is formatted like "A:n1,B:n2,C:n3,D:n4,E:n5".

(c) The successful configuration for GPT-4 (i.e., BSS$\widetilde{\text{C}}$, as temperature is specified elsewhere), which uses the buttons scenario, suggestive framing, summarized histories, and reinforced chain-of-thought reasoning.

[SYSTEM] You are a bandit algorithm in a room with 5 buttons labeled blue, green, red, yellow, purple. Each button is associated with a Bernoulli distribution with a fixed but unknown mean; the means for the buttons could be different. For each button, when you press it, you will get a reward that is sampled from the button's associated distribution. You have 10 time steps and, on each time step, you can choose any button and receive the reward. Your goal is to maximize the total reward over the 10 time steps.

At each time step, I will show you a summary of your past choices and rewards. Then you must make the next choice, which must be exactly one of blue, green, red, yellow, purple. Let's think step by step to make sure we make a good choice. You must provide your final answer within the tags <Answer>COLOR</Answer> where COLOR is one of blue, green, red, yellow, purple.

[USER] So far you have played 2 times with your past choices and rewards summarized as follows:
blue button: pressed 1 times with average reward 1.00
green button: pressed 1 times with average reward 0.00
red button: pressed 0 times
yellow button: pressed 0 times
purple button: pressed 0 times

Which button will you choose next? Remember, YOU MUST provide your final answer within the tags <Answer>COLOR</Answer> where COLOR is one of blue, green, red, yellow, purple. Let's think step by step to make sure we make a good choice.

## C Scatter plots and summary tables

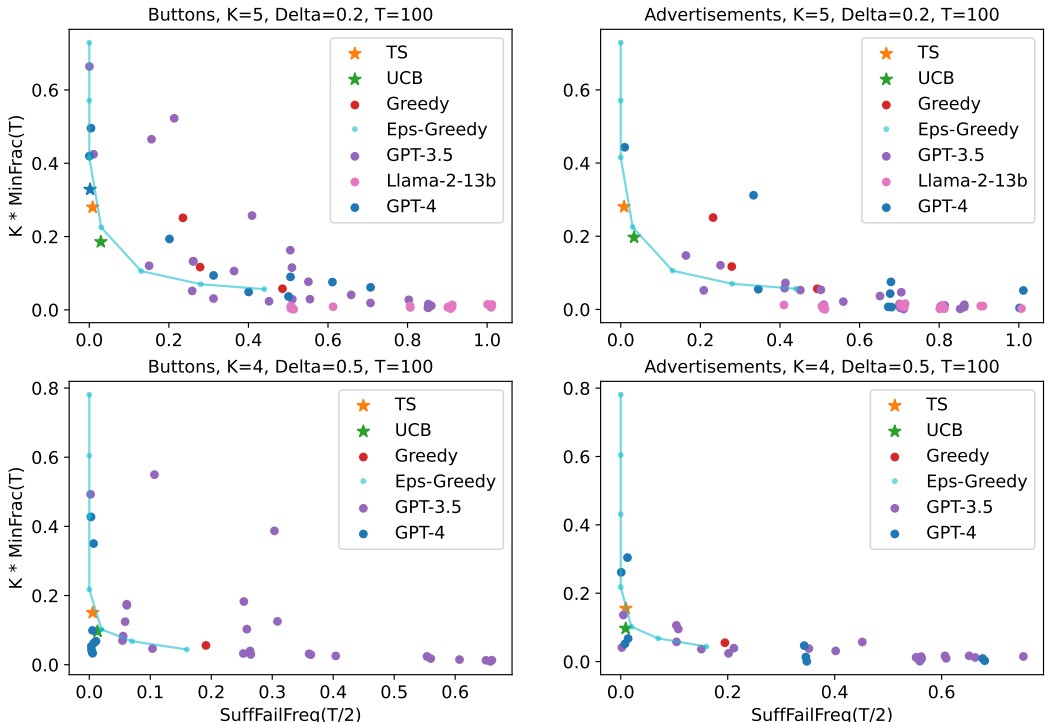

Figure 10: All **scatter plots** for the main experiments ($T = 100$): suffix failures vs. uniform-like failures. Specifically: $\texttt{SuffFailFreq}(T/2)$ vs $K \cdot \texttt{MinFrac}(T)$. Each LLM/configuration pair maps to a dot on this plane. (However, some dots may be hidden by some others.) We also plot $\epsilon$-GREEDY, tracing out the different tradeoffs obtained for different values of $\epsilon$.

**(a)** Hard MAB instance ($\Delta = 0.2$), buttons scenario, $N = 10$ replicates.

| | TS | UCB | Greedy | BNRN0 | BNRN1 | BNRND | BNRC0 | BNSN0 | BSRN0 | BSSC0 | BSSC1 | BSSCD | BSS$\bar{\text{C}}$0 |
|---|---|---|---|---|---|---|---|---|---|---|---|---|---|
| MedianReward | 0.47 | 0.55 | 0.40 | 0.63 | 0.70 | 0.33 | 0.35 | 0.60 | 0.45 | 0.68 | 0.28 | 0.37 | 0.47 |
| SuffFailFreq(T/2) | 0.01 | 0.02 | 0.48 | 0.50 | 0.40 | 0.00 | 0.50 | 0.60 | 0.70 | 0.30 | 0.20 | 0.00 | 0.00 |
| K*MinFrac | 0.28 | 0.18 | 0.05 | 0.03 | 0.04 | 0.41 | 0.09 | 0.07 | 0.05 | 0.09 | 0.19 | 0.49 | 0.33 |
| GreedyFrac | 0.62 | 0.76 | 1.00 | 0.52 | 0.46 | 0.45 | 0.78 | 0.99 | 0.59 | 0.93 | 0.88 | 0.49 | 0.69 |
| Replicates | 1000 | 1000 | 1000 | 10 | 10 | 10 | 10 | 10 | 10 | 10 | 10 | 10 | 10 |

**(b)** Hard MAB instance ($\Delta = 0.2$), advertisements scenario, $N = 3$ replicates.

| | TS | UCB | Greedy | ANRN0 | ANRN1 | ANRND | ANRC0 | ANSN0 | ASRN0 | ASSC0 | ASSC1 | ASSCD |
|---|---|---|---|---|---|---|---|---|---|---|---|---|
| MedianReward | 0.47 | 0.55 | 0.40 | 0.00 | -0.05 | -0.15 | 0.35 | 0.40 | 0.45 | 0.15 | 0.60 | -0.15 |
| SuffFailFreq(T/2) | 0.01 | 0.02 | 0.48 | 1.00 | 0.67 | 0.67 | 0.33 | 1.00 | 0.67 | 0.33 | 0.00 | 0.67 |
| K*MinFrac | 0.28 | 0.18 | 0.05 | 0.00 | 0.03 | 0.00 | 0.05 | 0.05 | 0.07 | 0.30 | 0.43 | 0.00 |
| GreedyFrac | 0.62 | 0.76 | 1.00 | 0.47 | 0.23 | 1.00 | 0.86 | 0.99 | 0.91 | 0.68 | 0.70 | 1.00 |
| Replicates | 1000 | 1000 | 1000 | 3 | 3 | 3 | 3 | 3 | 3 | 3 | 3 | 3 |

**(c)** Easy MAB instance ($\Delta = 0.5$), buttons scenario, $N = 3$ replicates.

| | TS | UCB | Greedy | BNRN0 | BNRN1 | BNRND | BNRC0 | BNSN0 | BSRN0 | BSSC0 | BSSC1 | BSSCD |
|---|---|---|---|---|---|---|---|---|---|---|---|---|
| MedianReward | 0.84 | 0.88 | 0.92 | 0.90 | 0.92 | 0.56 | 0.92 | 0.96 | 0.92 | 0.92 | 0.90 | 0.58 |
| SuffFailFreq(T/2) | 0.00 | 0.00 | 0.19 | 0.00 | 0.00 | 0.00 | 0.00 | 0.00 | 0.00 | 0.00 | 0.00 | 0.00 |
| K*MinFrac | 0.14 | 0.09 | 0.04 | 0.05 | 0.03 | 0.43 | 0.05 | 0.04 | 0.03 | 0.04 | 0.09 | 0.35 |
| GreedyFrac | 0.88 | 0.94 | 1.00 | 0.97 | 0.99 | 0.56 | 0.99 | 1.00 | 0.73 | 0.99 | 0.93 | 0.63 |
| Replicates | 1000 | 1000 | 1000 | 3 | 3 | 3 | 3 | 3 | 3 | 3 | 3 | 3 |

**(d)** Easy MAB instance ($\Delta = 0.5$), advertisements scenario, $N = 3$ replicates.

| | TS | UCB | Greedy | ANRN0 | ANRN1 | ANRND | ANRC0 | ANSN0 | ASRN0 | ASSC0 | ASSC1 | ASSCD |
|---|---|---|---|---|---|---|---|---|---|---|---|---|
| MedianReward | 0.84 | 0.88 | 0.92 | 0.88 | 0.88 | 0.08 | 0.88 | 0.90 | 0.88 | 0.70 | 0.68 | 0.08 |
| SuffFailFreq(T/2) | 0.00 | 0.00 | 0.19 | 0.33 | 0.33 | 0.67 | 0.00 | 0.33 | 0.00 | 0.00 | 0.00 | 0.67 |
| K*MinFrac | 0.14 | 0.09 | 0.04 | 0.01 | 0.00 | 0.00 | 0.04 | 0.04 | 0.07 | 0.25 | 0.29 | 0.00 |
| GreedyFrac | 0.88 | 0.94 | 1.00 | 0.81 | 0.95 | 1.00 | 0.94 | 1.00 | 0.96 | 0.81 | 0.73 | 1.00 |
| Replicates | 1000 | 1000 | 1000 | 3 | 3 | 3 | 3 | 3 | 3 | 3 | 3 | 3 |

Figure 11: GPT-4 for $T = 100$: the per-configuration **summary tables**. The "fails" row indicates that all replicates completed successfully.

| | MedianReward | SuffFailFreq(T/2) | K*MinFrac | GreedyFrac | Replicates |
|---|---|---|---|---|---|
| TS | 0.47 | 0.01 | 0.28 | 0.62 | 1000 |
| UCB | 0.55 | 0.02 | 0.18 | 0.76 | 1000 |
| Greedy | 0.40 | 0.48 | 0.05 | 1.00 | 1000 |
| BNRN0 | 0.22 | 0.50 | 0.16 | 0.30 | 20 |
| BNRN1 | 0.22 | 0.00 | 0.41 | 0.28 | 20 |
| BNRND | 0.12 | 0.55 | 0.07 | 0.40 | 20 |
| BNRC0 | 0.12 | 0.80 | 0.01 | 0.51 | 20 |
| BNRC1 | 0.10 | 0.50 | 0.03 | 0.57 | 20 |
| BNRCD | 0.65 | 0.45 | 0.01 | 0.75 | 20 |
| BNSN0 | 0.12 | 0.85 | 0.00 | 1.00 | 20 |
| BNSN1 | 0.22 | 0.25 | 0.04 | 0.76 | 20 |
| BNSND | 0.20 | 0.20 | 0.52 | 0.38 | 20 |
| BNSC0 | 0.12 | 0.85 | 0.00 | 0.95 | 20 |
| BNSC1 | 0.22 | 0.70 | 0.01 | 0.88 | 20 |
| BNSCD | 0.05 | 0.50 | 0.11 | 0.50 | 20 |
| BSRN0 | 0.17 | 0.30 | 0.25 | 0.32 | 20 |
| BSRN1 | 0.25 | 0.00 | 0.66 | 0.29 | 20 |
| BSRND | 0.42 | 0.25 | 0.12 | 0.33 | 20 |
| BSRC0 | 0.10 | 0.65 | 0.03 | 0.44 | 20 |
| BSRC1 | 0.05 | 0.25 | 0.12 | 0.47 | 20 |
| BSRCD | 0.28 | 0.15 | 0.11 | 0.60 | 20 |
| BSSN0 | 0.12 | 0.85 | 0.00 | 1.00 | 20 |
| BSSN1 | 0.25 | 0.30 | 0.03 | 0.78 | 20 |
| BSSND | 0.25 | 0.15 | 0.45 | 0.42 | 20 |
| BSSC0 | 0.17 | 0.85 | 0.00 | 1.00 | 20 |
| BSSC1 | 0.17 | 0.55 | 0.02 | 0.83 | 20 |
| BSSCD | 0.20 | 0.35 | 0.10 | 0.78 | 20 |

Figure 12: GPT-3.5 for $T = 100$: the per-configuration **summary table**. The buttons scenario, hard MAB instance.

|  | MedianReward | SuffFailFreq(T/2) | K*MinFrac | GreedyFrac | Replicates |
|---|---|---|---|---|---|
| TS | 0.47 | 0.01 | 0.28 | 0.62 | 1000 |
| UCB | 0.55 | 0.02 | 0.18 | 0.76 | 1000 |
| Greedy | 0.40 | 0.48 | 0.05 | 1.00 | 1000 |
| ANRN0 | 0.22 | 0.65 | 0.03 | 0.48 | 20 |
| ANRN1 | 0.22 | 0.50 | 0.05 | 0.33 | 20 |
| ANRND | 0.15 | 0.70 | 0.00 | 1.00 | 20 |
| ANRC0 | 0.15 | 0.85 | 0.00 | 0.98 | 20 |
| ANRC1 | 0.20 | 0.50 | 0.00 | 0.80 | 20 |
| ANRCD | 0.15 | 0.70 | 0.00 | 1.00 | 20 |
| ANSN0 | 0.12 | 0.85 | 0.00 | 1.00 | 20 |
| ANSN1 | 0.12 | 0.20 | 0.04 | 0.93 | 20 |
| ANSND | 0.15 | 0.70 | 0.00 | 1.00 | 20 |
| ANSC0 | 0.17 | 0.80 | 0.00 | 1.00 | 20 |
| ANSC1 | 0.12 | 0.55 | 0.01 | 0.93 | 20 |
| ANSCD | 0.15 | 0.70 | 0.00 | 1.00 | 20 |
| ASRN0 | 0.25 | 0.70 | 0.03 | 0.48 | 20 |
| ASRN1 | 0.05 | 0.42 | 0.06 | 0.28 | 20 |
| ASRND | 0.15 | 0.70 | 0.00 | 1.00 | 20 |
| ASRC0 | 0.37 | 0.40 | 0.06 | 0.64 | 20 |
| ASRC1 | 0.30 | 0.25 | 0.11 | 0.65 | 20 |
| ASRCD | 0.15 | 0.70 | 0.00 | 1.00 | 20 |
| ASSN0 | 0.15 | 0.85 | 0.00 | 1.00 | 20 |
| ASSN1 | 0.25 | 0.42 | 0.05 | 0.92 | 20 |
| ASSND | 0.15 | 0.70 | 0.00 | 1.00 | 20 |
| ASSC0 | 0.12 | 0.80 | 0.01 | 0.99 | 20 |
| ASSC1 | 0.30 | 0.15 | 0.14 | 0.83 | 20 |
| ASSCD | 0.15 | 0.70 | 0.00 | 1.00 | 20 |

Figure 13: GPT-3.5 for $T = 100$: the per-configuration **summary table**. The advertisements scenario, hard MAB instance.

| | MedianReward | SuffFailFreq(T/2) | K*MinFrac | GreedyFrac | Replicates |
|---|---|---|---|---|---|
| TS | 0.84 | 0.00 | 0.14 | 0.88 | 1000 |
| UCB | 0.88 | 0.00 | 0.09 | 0.94 | 1000 |
| Greedy | 0.92 | 0.19 | 0.04 | 1.00 | 1000 |
| BNRN0 | 0.23 | 0.55 | 0.02 | 0.85 | 20 |
| BNRN1 | 0.72 | 0.05 | 0.16 | 0.62 | 20 |
| BNRND | 0.14 | 0.25 | 0.17 | 0.46 | 20 |
| BNRC0 | 0.84 | 0.25 | 0.03 | 0.56 | 20 |
| BNRC1 | 0.81 | 0.05 | 0.08 | 0.77 | 20 |
| BNRCD | 0.88 | 0.10 | 0.04 | 0.92 | 20 |
| BNSN0 | 0.18 | 0.65 | 0.00 | 1.00 | 20 |
| BNSN1 | 0.60 | 0.40 | 0.02 | 0.89 | 20 |
| BNSND | 0.26 | 0.10 | 0.54 | 0.52 | 20 |
| BNSC0 | 0.18 | 0.65 | 0.00 | 1.00 | 20 |
| BNSC1 | 0.16 | 0.55 | 0.01 | 0.95 | 20 |
| BNSCD | 0.62 | 0.35 | 0.03 | 0.77 | 20 |
| BSRN0 | 0.73 | 0.30 | 0.11 | 0.57 | 20 |
| BSRN1 | 0.35 | 0.00 | 0.48 | 0.42 | 20 |
| BSRND | 0.21 | 0.25 | 0.09 | 0.43 | 20 |
| BSRC0 | 0.87 | 0.05 | 0.06 | 0.72 | 20 |
| BSRC1 | 0.73 | 0.05 | 0.16 | 0.72 | 20 |
| BSRCD | 0.81 | 0.05 | 0.11 | 0.76 | 20 |
| BSSN0 | 0.18 | 0.65 | 0.00 | 1.00 | 20 |
| BSSN1 | 0.17 | 0.25 | 0.02 | 0.89 | 20 |
| BSSND | 0.26 | 0.30 | 0.39 | 0.60 | 20 |
| BSSC0 | 0.19 | 0.60 | 0.00 | 0.99 | 20 |
| BSSC1 | 0.53 | 0.35 | 0.03 | 0.82 | 20 |
| BSSCD | 0.78 | 0.25 | 0.02 | 0.90 | 20 |

Figure 14: GPT-3.5 for $T = 100$: the per-configuration **summary table**. The buttons scenario, easy MAB instance.

|        | MedianReward | SuffFailFreq(T/2) | K*MinFrac | GreedyFrac | Replicates |
|--------|--------------|-------------------|-----------|------------|------------|
| TS     | 0.84         | 0.00              | 0.14      | 0.88       | 1000       |
| UCB    | 0.88         | 0.00              | 0.09      | 0.94       | 1000       |
| Greedy | 0.92         | 0.19              | 0.04      | 1.00       | 1000       |
| ANRN0  | 0.18         | 0.65              | 0.01      | 0.81       | 20         |
| ANRN1  | 0.10         | 0.35              | 0.03      | 0.47       | 20         |
| ANRND  | 0.10         | 0.55              | 0.00      | 1.00       | 20         |
| ANRC0  | 0.13         | 0.60              | 0.00      | 0.96       | 20         |
| ANRC1  | 0.77         | 0.35              | 0.03      | 0.89       | 20         |
| ANRCD  | 0.10         | 0.55              | 0.00      | 1.00       | 20         |
| ANSN0  | 0.18         | 0.65              | 0.00      | 1.00       | 20         |
| ANSN1  | 0.69         | 0.15              | 0.03      | 0.97       | 20         |
| ANSND  | 0.10         | 0.55              | 0.00      | 1.00       | 20         |
| ANSC0  | 0.23         | 0.60              | 0.00      | 1.00       | 20         |
| ANSC1  | 0.71         | 0.20              | 0.03      | 0.96       | 20         |
| ANSCD  | 0.10         | 0.55              | 0.00      | 1.00       | 20         |
| ASRN0  | 0.08         | 0.75              | 0.01      | 0.81       | 20         |
| ASRN1  | 0.08         | 0.45              | 0.05      | 0.40       | 20         |
| ASRND  | 0.10         | 0.55              | 0.00      | 1.00       | 20         |
| ASRC0  | 0.68         | 0.10              | 0.08      | 0.86       | 20         |
| ASRC1  | 0.74         | 0.00              | 0.13      | 0.86       | 20         |
| ASRCD  | 0.10         | 0.55              | 0.00      | 1.00       | 20         |
| ASSN0  | 0.29         | 0.00              | 0.04      | 0.92       | 20         |
| ASSN1  | 0.79         | 0.10              | 0.05      | 0.93       | 20         |
| ASSND  | 0.10         | 0.55              | 0.00      | 1.00       | 20         |
| ASSC0  | 0.89         | 0.20              | 0.01      | 1.00       | 20         |
| ASSC1  | 0.82         | 0.10              | 0.11      | 0.92       | 20         |
| ASSCD  | 0.10         | 0.55              | 0.00      | 1.00       | 20         |

Figure 15: GPT-3.5 for $T = 100$: the per-configuration **summary table**. The adverts scenario, easy MAB instance.

| | MedianReward | SuffFailFreq(T/2) | K*MinFrac | GreedyFrac | Replicates |
|---|---|---|---|---|---|
| TS | 0.47 | 0.01 | 0.28 | 0.62 | 1000 |
| UCB | 0.55 | 0.02 | 0.18 | 0.76 | 1000 |
| Greedy | 0.40 | 0.48 | 0.05 | 1.00 | 1000 |
| BNRN0 | -0.05 | 0.90 | 0.00 | 1.00 | 10 |
| BNRN1 | 0.07 | 0.90 | 0.00 | 1.00 | 10 |
| BNRC0 | 0.10 | 0.80 | 0.01 | 0.62 | 10 |
| BNRC1 | 0.28 | 0.90 | 0.00 | 0.89 | 10 |
| BNSN0 | 0.60 | 0.50 | 0.00 | 1.00 | 10 |
| BNSN1 | 0.60 | 0.50 | 0.00 | 1.00 | 10 |
| BNSC0 | 0.07 | 1.00 | 0.00 | 1.00 | 10 |
| BNSC1 | 0.47 | 0.60 | 0.00 | 1.00 | 10 |
| BSRN0 | -0.03 | 0.90 | 0.00 | 1.00 | 10 |
| BSRN1 | -0.08 | 1.00 | 0.00 | 0.93 | 10 |
| BSRC0 | 0.10 | 0.80 | 0.01 | 0.72 | 10 |
| BSRC1 | -0.08 | 1.00 | 0.01 | 0.67 | 10 |
| BSSN0 | 0.60 | 0.50 | 0.00 | 1.00 | 10 |
| BSSN1 | 0.60 | 0.50 | 0.00 | 1.00 | 10 |
| BSSC0 | 0.07 | 1.00 | 0.00 | 1.00 | 10 |
| BSSC1 | 0.22 | 0.90 | 0.00 | 1.00 | 10 |

Figure 16: LLAMA2 for $T = 100$: the per-configuration **summary tables**. The buttons scenario, hard MAB instance.

| | MedianReward | SuffFailFreq(T/2) | K*MinFrac | GreedyFrac | Replicates |
|---|---|---|---|---|---|
| TS | 0.47 | 0.01 | 0.28 | 0.62 | 1000 |
| UCB | 0.55 | 0.02 | 0.18 | 0.76 | 1000 |
| Greedy | 0.40 | 0.48 | 0.05 | 1.00 | 1000 |
| BNRN0 | -0.05 | 0.90 | 0.00 | 1.00 | 10 |
| BNRN1 | 0.07 | 0.90 | 0.00 | 1.00 | 10 |
| BNRC0 | 0.10 | 0.80 | 0.01 | 0.62 | 10 |
| BNRC1 | 0.28 | 0.90 | 0.00 | 0.89 | 10 |
| BNSN0 | 0.60 | 0.50 | 0.00 | 1.00 | 10 |
| BNSN1 | 0.60 | 0.50 | 0.00 | 1.00 | 10 |
| BNSC0 | 0.07 | 1.00 | 0.00 | 1.00 | 10 |
| BNSC1 | 0.47 | 0.60 | 0.00 | 1.00 | 10 |
| BSRN0 | -0.03 | 0.90 | 0.00 | 1.00 | 10 |
| BSRN1 | -0.08 | 1.00 | 0.00 | 0.93 | 10 |
| BSRC0 | 0.10 | 0.80 | 0.01 | 0.72 | 10 |
| BSRC1 | -0.08 | 1.00 | 0.01 | 0.67 | 10 |
| BSSN0 | 0.60 | 0.50 | 0.00 | 1.00 | 10 |
| BSSN1 | 0.60 | 0.50 | 0.00 | 1.00 | 10 |
| BSSC0 | 0.07 | 1.00 | 0.00 | 1.00 | 10 |
| BSSC1 | 0.22 | 0.90 | 0.00 | 1.00 | 10 |

Figure 17: LLAMA2 for $T = 100$: the per-configuration **summary tables**. The advertisements scenario, hard MAB instance.

# D  Investigating successes: additional visualization for Subsection 3.2.

We provide an additional visualization for Subsection 3.2, with some qualitative evidence toward the success of BSSC̃0, as well as the failure of other configurations. In Figure 18 we visualize the arm chosen at each time step for various replicates of several different methods (LLMs and baselines). Specifically, we have four replicates for the basic configuration (BNRN0) and the two configurations with reinforced CoT (BSRC̃0 and BSSC̃0), as well as one replicate of each of the baseline algorithms. We see that the basic configuration BNRN0 tends to commit to a single arm for several rounds, a behavior that is similar to that of GREEDY and very different from both UCB and TS. BSRC̃0 also commits for long periods, but to a lesser extent than the basic configuration. In contrast, BSSC̃0 switches arms much more frequently, and qualitatively appears much more similar to TS.

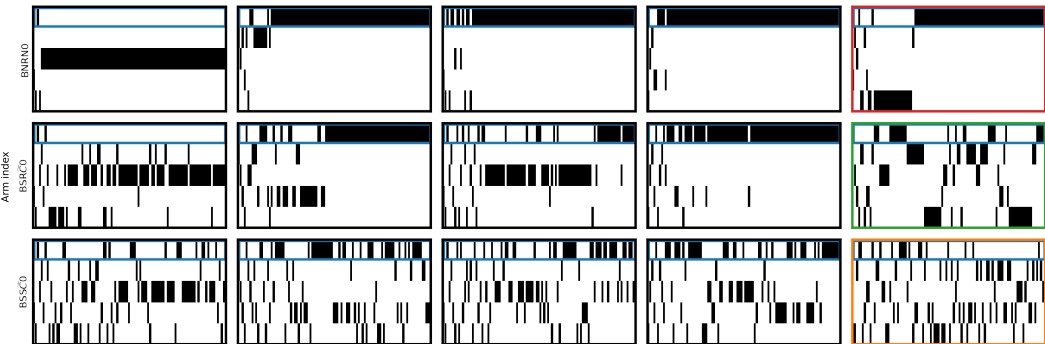

Figure 18: Traces of the arm chosen at each time step for (a) 4 of the replicates of the basic configuration (GPT-4-BNRN0) (left four cells in top row), (b) 4 of the replicates of GPT-4-BSRC̃0 (left four cells of the middle row), (c) 4 of the replicates of GPT-4-BSSC̃0 (left four cells of the bottom row), as well as one replicate of GREEDY (red border), UCB (green border) and TS (orange border). For each of the $T = 100$ time steps (X-axis) we indicate which of the five arms was chosen (Y-axis). The best arm is the top row of each plot, highlighted with blue boxes.

# E  Root causes

Our experimental findings above shed light on how LLM-based decision making agents behave, but it is also worthwhile to understand *why* they behave the way they do (and particularly, why they fail). This question is rather challenging to answer decisively, but two natural hypotheses are that the configurations we consider (outside of GPT-4-BSS$\widetilde{\text{C}}$0) are either a) too greedy, or b) too uniform-like. In this section, we describe how our experiments offer some insight into this hypotheses.

|  | GreedyFrac | | | LeastFrac | | |
|---|---|---|---|---|---|---|
| TS | 0.60 | 0.54 | 0.53 | 0.30 | 0.12 | 0.12 |
| UCB | 0.84 | 0.66 | 0.55 | 0.46 | 0.09 | 0.26 |
| BNRN0 | 0.34 | 0.36 | 0.50 | 0.30 | 0.30 | 0.24 |
| BNRC0 | 0.50 | 0.84 | 0.58 | 0.12 | 0 | 0.04 |
| BNSN0 | 0.82 | 0.94 | 0.84 | 0.28 | 0 | 0 |
| BSRN0 | 0.20 | 0.18 | 0.22 | 0.60 | 0.38 | 0.38 |
| Data source | Unif | UCB | TS | Unif | UCB | TS |

Figure 19: Per-round decisions with some GPT-3.5 configurations. $T = 100$, histories of length $t = 30$, hard MAB instance.

First, focusing on GPT-4, our experiments reveal qualitatively different behavior between the easy and hard instances (Figure 11(a) and Figure 11(c)). Indeed, the easy instance appears to be *much* easier; most GPT-4 configurations avoid suffix failures and accrue large rewards on this instance, and the GreedyFrac statistic offers a potential explanation as to why. On the easy instance, most GPT-4 configurations have very high GreedyFrac values, so they behave similarly to GREEDY, which performs quite well (even though GREEDY provably fails with small constant probability and, empirically, has many suffix failures on this instance).[10] A plausible hypothesis from this is that GPT-4 performs quite well in low-noise settings, which is precisely when GREEDY also performs well.

A stronger hypothesis would be that most GPT-4 configurations (except perhaps those using reinforced CoT) behave like GREEDY on *all* instances, but this hypothesis is invalidated by the GreedyFrac statistics for our experiments on the hard instance. On the hard instance, it seems that most GPT-4 configurations are doing something non-trivial (albeit flawed); their behavior is neither completely GREEDY-like nor like uniform-at-random.

Toward a more fine-grained understanding, we ran a collection of small-scale secondary experiments focusing on the *per-round decisions* of LLM-agents. The experiments focus on a single round $t$ in a bandit problem. Each experiment considers a particular "data source" (a distribution of bandit histories), samples $N = 50$ bandit histories of length $t$ from this distribution, and presents them to the agents (the LLMs and the baselines) and asks them to output an arm or distribution over arms. We track two statistics for each agent: GreedyFrac and LeastFrac, the fraction of replicates in which the agent chose, resp., an empirically best arm so far and a least-chosen arm so far. We vary the data source, i.e., the algorithm which generates the history. In particular, we consider histories generated by sampling uniformly at random (Unif) and by running our baselines UCB and TS for $t$ rounds.

Results are summarized in Figure 19. Unfortunately, we find that per-round performance of both the LLMs and the baselines is very sensitive to the particular data source. For example, the MinFrac statistic of UCB can vary from as high as 0.46 on histories generated uniformly at random to as low as 0.09 on histories generated by UCB itself. It seems plausible to conclude the BNSN0 is too greedy while BSRN0 is too uniform, but the statistics for the other two LLM configurations (BNRN0 and BNRC0)—both of which fail in our longitudinal experiments—fall within the reasonable range provided by the baselines. Thus, we find that it is challenging to assess whether LLM agents are too greedy or too uniform-like based on per-round decisions, even though these agents behave rather differently from the baselines in the longitudinal experiments.

---

[10]Indeed, in Figure 11(c) we see that most GPT-4 configurations have very high GreedyFrac but no suffix failures. Apparently, even a very small amount of exploration suffices for easy instances (and makes a big difference, relative to GREEDY). However, this should not be construed as evidence for the more general and robust exploratory behavior necessary for harder bandit instances.

