# OpenReview forum: "Can large language models explore in-context?"
_NeurIPS.cc/2024/Conference — NeurIPS 2024 poster_

### Official Review · Reviewer_Jht5 · 2024-07-13

**Soundness:** 3
**Presentation:** 3
**Contribution:** 2
**Rating:** 5
**Confidence:** 4

**Summary:**

In this study, the authors investigate the impact of diverse prompt designs, including environmental descriptions, summarizing interaction history, and chain of thought reasoning, on the exploration behavior of three distinct LLMs in a multi-armed bandit task. The results demonstrated that none of the models engaged in exploration without substantial interventions. The only configuration that demonstrated efficacy was the GPT-4 model, which was prompted with change of thought reasoning and an externally summarised interaction history. The authors demonstrate that the typical failure mode for these models is either suffix errors, wherein the LLM fails to select the optimal arm even once in the initial stages, or uniform sampling of all options. In conclusion, the authors raise concerns about the external validity of the working configuration, noting that it may not be scalable to more complex task settings. They also speculate on the potential algorithmic interventions that could be employed to encourage LLMs to explore in such settings.

**Strengths:**

Whether or not LLMs can explored in context in a reinforcement learning task and if so, how is a question worth investigating and approach to study it including the paradigm used are tried and tested.

**Weaknesses:**

The research question and paradigm used to study that question have been previously investigated in the literature [1-5]. However, in contrast to previous studies, the authors have conducted a more comprehensive search of prompt designs, included a greater number of options in the multi-armed bandit task, and analyzed the failure modes of the models using two novel measures. Nevertheless, I have significant concerns regarding the work's novelty and its contribution's strength. While the results are interesting, the main contribution of the work is somewhat limited, as the subset of evaluations performed has already been studied by previous works (see questions and limitations for details).

Additionally, the authors have not adequately cited related works [1-8].

[1] Binz, M., & Schulz, E. (2023). Using cognitive psychology to understand GPT-3. _Proceedings of the National Academy of Sciences_, _120_(6), e2218523120.

[2] Schubert, J. A., Jagadish, A. K., Binz, M., & Schulz, E. (2024). In-context learning agents are asymmetric belief updaters. _arXiv preprint arXiv:2402.03969_.

[3] Hayes, W. M., Yax, N., & Palminteri, S. (2024). Large Language Models are Biased Reinforcement Learners. _arXiv preprint arXiv:2405.11422_.

[4] Coda-Forno, J., Binz, M., Akata, Z., Botvinick, M., Wang, J., & Schulz, E. (2023). Meta-in-context learning in large language models. _Advances in Neural Information Processing Systems_, _36_, 65189-65201.

[5] William M Hayes, Nicolas Yax, and Stefano Palminteri. Relative value biases in large language models. arXiv preprint arXiv:2401.14530, 2024.

[6] Julian Coda-Forno, Marcel Binz, Jane X Wang, and Eric Schulz. Cogbench: a large language model walks into a psychology lab. arXiv preprint arXiv:2402.18225, 2024

[7] Thilo Hagendorff. Machine psychology: Investigating emergent capabilities and behavior in large language models using psychological methods.

[8] Lampinen, A. K., Dasgupta, I., Chan, S. C., Matthewson, K., Tessler, M. H., Creswell, A., ... & Hill, F. (2022). Can language models learn from explanations in context?. _arXiv preprint arXiv:2204.02329_.

**Questions:**

What factors within the model (size, architecture, RLHF, internal representation), its training dataset, and training protocol (at a specific point during training) could be preventing LLMs to fail at exploration? It may be beneficial for the authors to consider investigating some of these factors further, which could potentially enhance the quality of the paper. It would be beneficial to investigate the emergence of exploration behavior in LLMs by examining smaller models, such as GPT-2/LLaMa-2.

It has been demonstrated that LLM models can be fine-tuned for reinforcement learning tasks, enabling them to function as decision-makers capable of generalizing to novel tasks with high efficiency [1-2]. These findings suggest that, despite their limited capacity for exploration when tested off-the-shelf, LLMs can be effectively fine-tuned to enhance their ability to explore. It would be valuable for the authors to investigate whether fine-tuning LLMs (such as the GPT-2 /LLaMa-2 model) on simple RL tasks could facilitate their exploration capabilities.

[1] Cao, Y., Zhao, H., Cheng, Y., Shu, T., Liu, G., Liang, G., ... & Li, Y. (2024). Survey on large language model-enhanced reinforcement learning: Concept, taxonomy, and methods. _arXiv preprint arXiv:2404.00282_.

[3] T. Carta, C. Romac, T. Wolf, S. Lamprier, O. Sigaud, and P.-Y. Oudeyer, “Grounding Large Language Models in Interactive Environments with Online Reinforcement Learning,” Sep. 2023

**Limitations:**

The failure mode of an agent following an epsilon greedy policy with a high epsilon value is analogous to that of Suffix Failure. Conversely, KMinFrac is analogous to the failure mode of an agent following a random policy or an epsilon greedy policy with a low epsilon value. Therefore,  I believe the metrics themselves are quite trivial and do not constitute a novel contribution.

Rather than relying on a qualitative comparison between LLM's choices and those of alternative bandit algorithms, the authors may find it beneficial to fit the models of exploration to LLM's choices and conduct a more quantitative evaluation [1]

[1] Gershman, S. J. Deconstructing the human algorithms for exploration. Cognition, 173:34–42, 2018.

---

> ### Author Rebuttal · Authors · 2024-08-06
>
> > *Additionally, the authors have not adequately cited related works [1-8].*
>
> Thanks for the references! We actually became aware of some of these between the submission and now and have already incorporated appropriate discussions into the manuscript. But we’ll be sure to discuss them in a revision.
>
> Let us comment on [1-8], as per your list.
>
> [2,3,5,6] are concurrent, unpublished works (from 2024), and should not be considered as novelty issues. (Note that our paper has appeared on arxiv since March 2024.) We’ll provide a detailed comparison in a revision*.
>
> Of the remaining papers, [1,4] consider bandit tasks at a very small scale (T=10, 2 arms), which is really insufficient to distinguish between good and bad algorithms, and [7,8] have no bandit tasks at all.
>
> All bandit tasks but [4] focus on LLM vs human comparisons, whereas we focus on LLM vs algorithms. We believe this is an important distinction.
>
> *Two brief notes, though: [2] only experiments with Claude v1.3, which is considered roughly on the level of GPT-3.5 and hence probably not indicative of GPT-4 performance;  [5] only considers bandit tasks at a very small scale.
>
> > [Q1] *What factors within the model (size, architecture, RLHF, internal representation), its training dataset, and training protocol (at a specific point during training) could be preventing LLMs to fail at exploration? It may be beneficial for the authors to consider investigating some of these factors further, which could potentially enhance the quality of the paper. It would be beneficial to investigate the emergence of exploration behavior in LLMs by examining smaller models, such as GPT-2/LLaMa-2.*
>
> Thanks for the suggestion. We did experiment with LLaMA-2 in the paper (results are summarized in Fig 3 and reported with more details in the appendix) and found that it is much worse than GPT-3.5-Turbo, which itself is much worse than GPT-4. In a sense, this already demonstrates that exploration behavior is emergent, and that it perhaps can be improved further by scaling.
>
> > [Q2] *It has been demonstrated that LLM models can be fine-tuned for reinforcement learning tasks, enabling them to function as decision-makers capable of generalizing to novel tasks with high efficiency [1-2]. These findings suggest that, despite their limited capacity for exploration when tested off-the-shelf, LLMs can be effectively fine-tuned to enhance their ability to explore. It would be valuable for the authors to investigate whether fine-tuning LLMs (such as the GPT-2 /LLaMa-2 model) on simple RL tasks could facilitate their exploration capabilities.*
>
> Thanks for these references. In our related work section (Appendix A), we discussed some works that train language models from scratch on trajectory data from another agent, resulting in suitable decision-making (and exploratory) behavior. Based on this, it’s not too surprising that LLMs can be fine-tuned to improve their decision-making capabilities; we’ll definitely include the citations. However, it’s worth emphasizing that fine-tuning requires expertise & money and/or infrastructure & access.  Many practitioners are using (or will use) off-the-shelf LLMs for decision-making tasks without fine-tuning, and so it is critical to understand what is possible with standard training.
>
> > *The failure mode of an agent following an epsilon greedy policy with a high epsilon value is analogous to that of Suffix Failure. Conversely, KMinFrac is analogous to the failure mode of an agent following a random policy or an epsilon greedy policy with a low epsilon value. Therefore, I believe the metrics themselves are quite trivial and do not constitute a novel contribution.
> Rather than relying on a qualitative comparison between LLM's choices and those of alternative bandit algorithms, the authors may find it beneficial to fit the models of exploration to LLM's choices and conduct a more quantitative evaluation*
>
> Both approaches seem perfectly reasonable to us but we respectfully disagree that one is “qualitative” while the other is “quantitative.” Our approach is completely quantitative: we suggest quantitative statistics to measure, demonstrate that they correlate strongly with performance (measured via reward over a much longer time scale), and quantitatively compare to state-of-the-art methods. Compared with the suggested modeling approach, our statistics allow for direct measurement of exploratory behavior without imposing any modeling assumptions (which would likely be wrong/erroneous given the erratic behavior of LLMs that we witnessed in our experiments).

---

> ### Comment · Reviewer_Jht5 · 2024-08-12
> **Response to rebuttal**
>
> Thank the authors for their detailed responses to our queries.
>
> Re. Q1: Can the authors include a mixed-effects regression analysis, where they perform a regression from the abovementioned factors onto the surrogate measures to see how much each factor determines the extent of exploration and lack thereof? This allows us to understand the contributions of the factors at a more fine-grained level.
>
> Re. Q2: We agree with the authors that expertise & money and/or infrastructure prevent fine-tuning to be a plausible option for everyone. But we differ from them in that there is a sufficient audience who would be interested if there is a simple fine-tuning protocol that makes LLMs explore optimally.  We think smaller models can be a good test bed for such approaches and are quite confident that the protocol can even be standardized if shown to provide sufficient gains.
>
> Re. external validity concerns as raised by two other reviewers:
> We agree with the authors that the bandit task provides a well-studied, controlled setting to study exploration in LLMs.
> However, given that these models are typically made to solve tasks vastly different from the bandit setting, We still have concerns about whether the findings will hold for those settings. If authors find the same effects in coding or some problem-solving tasks with the help of their prompting strategy, that would greatly increase the potential impact of the findings.
>
> Keeping these points in mind, I am happy to increase my score from 3 to 5.

---

> > ### Author Response · Authors · 2024-08-12
> >
> > Thanks for the helpful feedback. We would be happy to include such a mixed-effects regression analysis in our next revision.

---

### Official Review · Reviewer_qwyY · 2024-07-13

**Soundness:** 2
**Presentation:** 3
**Contribution:** 2
**Rating:** 4
**Confidence:** 3

**Summary:**

This paper investigates whether large language models (LLMs) like GPT-3.5, GPT-4, and LLAMA2 can perform exploration in reinforcement learning settings without additional training. The authors focus on in-context learning, where the environment description and interaction history are provided entirely within the LLM prompt. They evaluate LLMs on multi-armed bandit problems, comparing their performance to standard bandit algorithms like Thompson Sampling (TS) and Upper Confidence Bound (UCB).

The key findings are: (1) Most LLM configurations fail to robustly explore, exhibiting either "suffix failures" (converging to a suboptimal arm) or "uniform-like failures" (selecting arms uniformly at random). (2) Only one configuration (GPT-4 with chain-of-thought reasoning and externally summarized history) showed satisfactory exploratory behavior comparable to TS. (3) LLM performance is highly sensitive to prompt design and varies significantly across model versions.

The authors conclude that non-trivial algorithmic interventions may be necessary for LLMs to function effectively as decision-making agents in complex settings.

**Strengths:**

-  The study uses multiple LLMs, various prompt designs, and compares against established baselines. The use of both easy and hard multi-armed bandit instances provides a comprehensive evaluation.
- The introduction of "suffix failure frequency" and "MinFrac" as surrogate statistics for identifying exploration failures is innovative and allows for more efficient evaluation of LLM performance.
- The work addresses an important question in the rapidly evolving field of LLM capabilities, with implications for using LLMs in decision-making tasks.

**Weaknesses:**

- While the focus on multi-armed bandits provides a clean experimental setup, it may not fully represent the challenges of more complex reinforcement learning problems. The generalizability of the findings to broader settings is unclear.
-  The paper is predominantly empirical. A theoretical framework for understanding why LLMs struggle with exploration could provide deeper insights and guide future research.
- While the authors attempt to explore reasons for LLM failures in Section 3.3, this analysis feels underdeveloped. A more systematic investigation of failure modes could yield valuable insights.
- While the authors compare summarized vs. raw history, more systematic ablations of different prompt components could help isolate the factors contributing to successful exploration.

**Questions:**

None.

**Limitations:**

Yes.

---

> ### Author Rebuttal · Authors · 2024-08-06
>
> > *While the focus on multi-armed bandits provides a clean experimental setup, it may not fully represent the challenges of more complex reinforcement learning problems. The generalizability of the findings to broader settings is unclear.*
>
> Failures on a fundamental special case such as MAB plausibly carry over to the broader scenarios. But we completely agree for the positive findings. However, this would be the case for any experiment setup: the findings do not necessarily generalize beyond the precise experiment setting. Among all possible experiment setups, we argue that simple/clean/fundamental special cases are preferable. MAB is precisely such a setup.
>
> > *The paper is predominantly empirical. A theoretical framework for understanding why LLMs struggle with exploration could provide deeper insights and guide future research.*
>
> Indeed! But we view it as a next step in this line of research. Our contribution is the prerequisite to this: we identify that there is a need for such a framework. Nevertheless, we hope that the suffix failure and MinFrac statistics provide some hint as to how one might develop a deeper understanding.
>
> > *While the authors attempt to explore reasons for LLM failures in Section 3.3, this analysis feels underdeveloped. A more systematic investigation of failure modes could yield valuable insights.*
>
> Agreed! We included section 3.3 primarily to point out the difficulty of such investigation, particularly of using  single-round experiments to isolate long-horizon issues. We can emphasize this in a revision.
>
> > *While the authors compare summarized vs. raw history, more systematic ablations of different prompt components could help isolate the factors contributing to successful exploration.*
>
> Again we agree. For this we were primarily limited by cost for GPT-4. Note that we have reported results for all 24 prompt configurations for GPT-3.5-Turbo in the appendix, so “systematic ablations” on a weaker LLM are included in the manuscript.

---

### Official Review · Reviewer_yw69 · 2024-07-15

**Soundness:** 2
**Presentation:** 3
**Contribution:** 2
**Rating:** 5
**Confidence:** 3

**Summary:**

This paper investigates the exploration capabilities of contemporary large language models. The authors use LLMs as agents in multi-armed bandit environments, describing the environment and interaction history in-context without any training interventions. Their experiments involve a variety of configurations and models, and they conclude that only GPT-4 with chain-of-thought reasoning and an externally summarized interaction history demonstrates satisfactory exploratory behavior.

**Strengths:**

- The paper comprehensively analyzes the LLMs in various configurations and prompt choices to understand their exploratory capabilities.

**Weaknesses:**

- In line 161, it is mentioned that no parameter tuning is performed for the baselines with tunable parameters. In this case, we are not sure that the baselines are performing to their highest capacity and therefore, the comparison might not be fair.
- Longer horizons T and replicates N might be required to obtain statistically significant results.

**Questions:**

- Why isn't parameter tuning performed for the baselines with tunable parameters?
- In line 97, it is mentioned that all the arms, except for the best arm, have the same mean reward. What is the reason for this choice?
- If the summary is not available for more complex situations, can few-shot examples help instead?

**Limitations:**

- Longer horizons T and replicates N might be required to obtain statistically significant results.

---

> ### Author Rebuttal · Authors · 2024-08-06
>
> > *In line 161, it is mentioned that no parameter tuning is performed for the baselines with tunable parameters. In this case, we are not sure that the baselines are performing to their highest capacity and therefore, the comparison might not be fair.”* (Also, Q1)
>
> This is rather standard in bandit experiments. It puts the baselines at a disadvantage, and even then they (for the most part) outperform the LLMs. So even without parameter tuning there is strong evidence that LLMs do not effectively explore.
>
> > *Longer horizons T and replicates N might be required to obtain statistically significant results.*
>
> This is a fair concern. As we discuss in Section 2 ("Scale of the experiments"), our scale (T x N x #configs x #instances) is near-prohibitive given the high cost of invoking modern LLMs, yet insufficient if one relies on standard statistics such as accumulated rewards. This necessitates a more subtle study focusing on surrogate statistics focusing suffix and uniform-like failures.
>
> We note that most/all follow-up or concurrent studies of LLMs in bandit problems were smaller-scale (e.g. [68]; [1-6] mentioned by Reviewer Jht5; Wu et al, "A benchmark for LLMs as intelligent agents".ICLR'24; Park et al., "Do LLM agents have regret?A case study in online learning and games. arXiv:2403.16843, 2024).
>
> > [Q2] *In line 97, it is mentioned that “all the arms, except for the best arm, have the same mean reward”. What is the reason for this choice?*
>
> This choice (a) is the standard lower bound construction for MABs, and (b) has the fewest degrees of freedom, leading to a simpler and more defensible experiment setup. Of course other designs are also reasonable.
>
> > [Q3] *If the summary is not available for more complex situations, can few-shot examples help instead?*
>
> We suspect so, and mention few-shot prompting as a natural direction for future work in Section 4.

---

### Official Review · Reviewer_6C6Y · 2024-07-16

**Soundness:** 3
**Presentation:** 3
**Contribution:** 3
**Rating:** 7
**Confidence:** 4

**Summary:**

This paper investigates whether popular LLMs can engage in exploration in a in-context manner (all experiences are stored as context/prompt). To achieve, the paper deploys LLMs (GPT-3.5, GPT-4, LLAMA2) as agents in multi-armed bandit environments, using various prompt designs to specify the environment description and interaction history entirely in-context. The results show that only GPT-4 with chain-of-thought reasoning and externally summarized interaction history exhibited satisfactory exploratory behavior. Other configurations failed to robustly explore. The authors suggest that non-trivial algorithmic interventions, such as fine-tuning or dataset curation, may be required for LLMs to effectively explore in more complex settings.

**Strengths:**

The paper investigates an interesting question: can LLMs explore in-context. Human holds the ability to explore in-context, with their self-summarized histories or abstractions. The results may indicate the intelligence embedded in LLMs is still not aligned with human.

The findings have meaningful implications for the use of LLMs in decision-making tasks (especially when directly employing LLM as the agent). The identification of the need for external summarization and potential training interventions makes sense.

**Weaknesses:**

As an experimental article, the findings are naive and obvious: the LLMs are not designed for solving decision-making tasks. It is consistent with intuition that LLMs can explore in-context when CoT with summarized history.

I feel that experiment results on more challenging tasks are necessary, such as some textual interactive games. In practical, the scenarios human encounters and explores are more complex than a bandit.

More discussions about related work are appreciated.

There lacks a more systematic analysis of which specific prompt design elements are most critical for successful exploration? This could help in understanding the sensitivity of LLMs to prompt variations.

**Questions:**

1. Only one configuration (GPT-4 with chain-of-thought reasoning and an externally summarized interaction history) resulted in satisfactory exploratory behavior. Why this specific configuration was successful and others were not? Are there any insights into the underlying mechanisms?
2. The paper concludes that external summarization is essential for desirable LLM behavior in simple environments. How do the authors envision scaling this approach to more complex settings where external summarization might be infeasible? Are there any preliminary ideas or experiments that could address this limitation?

**Limitations:**

I do not find any limitations discussed in the main body.

---

> ### Author Rebuttal · Authors · 2024-08-06
>
> > *As an experimental article, the findings are naive and obvious: the LLMs are not designed for solving decision-making tasks. It is consistent with intuition that LLMs can explore in-context when CoT with summarized history.*
>
> Whether LLMs are “designed” for decision-making tasks is not very relevant to our study. LLMs are already being deployed as decision-making agents (e.g. [40, 54, 63, 48, 5, 71]), and so it is crucial to understand their capabilities as such. Besides, LLMs are not explicitly “designed” for many tasks that they are good at.
>
> We do not understand how LLMs not being designed for (or failing at) decision-making is consistent with the success of CoT and history summarization. Indeed, it was unclear if CoT would help much and LLMs are quite capable of performing the simple arithmetic required for history summarization.)
>
> Re. “naive and obvious”: we emphasize that the scale of our experiments (as discussed in Section 2) was already near-prohibitive given the LLM costs, but still insufficient with the standard statistics (cumulative rewards), and hence necessitated a considerable subtlety (e.g., tracking suffix and uniform-like failures). The “prompt space” also required a somewhat careful design.
>
> > *I feel that experiment results on more challenging tasks are necessary, such as some textual interactive games. In practical, the scenarios human encounters and explores are more complex than a bandit.*
>
> Real-world decision-making problems are indeed much more complicated than ours. However we emphasize that our MAB setting distills the essence of exploration (as justified by decades of theoretical research) and embeds as a special case into essentially all RL/decision-making formulations. If an agent cannot solve a simple task, what hope does it have for more complex ones?
>
> Also: simpler settings require the experimenter to make fewer arbitrary choices, and can lead to more generalizable findings.
>
> > *More discussions about related work are appreciated.*
>
> You might have missed our detailed related work section, deferred to Appendix A due to page limit. Nevertheless, we’d be happy to add any specific references you’d like.
>
> > *There lacks a more systematic analysis of which specific prompt design elements are most critical for successful exploration? This could help in understanding the sensitivity of LLMs to prompt variations.*
>
> We already consider a non-trivial space of prompt designs, see Fig2. (And we note that it is at least comparable to that considered in concurrent work on bandit tasks for LLMs.)
>
> We totally agree a more systematic analysis of prompt designs could be very useful. However, it is also extremely challenging because the prompt space (even just for simple bandit problems) is already massive. It might be more tractable to do this in simpler (non-sequential) settings, but LLM sensitivity suggests it is difficult to generalize the findings.
>
> > [Q1] *Only one configuration (GPT-4 with chain-of-thought reasoning and an externally summarized interaction history) resulted in satisfactory exploratory behavior. Why this specific configuration was successful and others were not? Are there any insights into the underlying mechanisms?*
>
> An advanced LLM (GPT-4) and summarization are obvious benefits. The latter avoids occasional arithmetic errors that LLMs are known to make (and we found in our logs). CoT is known to help in many other tasks, but the reason is unobvious. One hypothesis (from other work) is that CoT gives the model access to more compute, which improves reasoning capabilities. In our experiment logs,  CoTs often did describe correct algorithm behavior, like the correct posterior or upper confidence bound computations. We can describe this in more detail in the final version.
>
> > [Q2] *The paper concludes that external summarization is essential for desirable LLM behavior in simple environments. How do the authors envision scaling this approach to more complex settings where external summarization might be infeasible? Are there any preliminary ideas or experiments that could address this limitation?*
>
> This is a fundamental open question for future work. One perhaps naive idea is via “orchestration”: we invoke the LLM once and ask it to summarize the history, then we use this “self-produced” summarization in a second invocation where we request a decision. While not provably correct, such summaries seem simple enough to experiment with.
>
> > *The results may indicate the intelligence embedded in LLMs is still not aligned with human.*
>
> True. However, we focus on comparing LLMs to algorithms rather than humans.

---

> > ### Comment · Reviewer_6C6Y · 2024-08-08
> > **Response to rebuttal**
> >
> > It seems that major concerns raised by other reviewers are the complexity of the evaluation tasks. While I expect to see the performance of LLM in-context exploration in challenging tasks, I agree with the authors that MAB setting presents significance in exploration. Exploration in simple tasks is the foundation for more complicated tasks.
> >
> > Overall, I feel that LLM in-context exploration is an interesting direction. Thus I raise my score from 6 to 7.

---

### Author Rebuttal · Authors · 2024-08-06

Thanks for taking the time to review our submission. To summarize our contributions, we perform a systematic analysis of the extent to which LLMs are capable of exploration, a core component of reinforcement learning/decision-making, by deploying LLMs as agents in multi-armed bandit environments. Out of all configurations we tried, we found that only one configuration resulted in satisfactory exploratory behavior. Along the way, we develop several statistics which allow us to measure the degree to which LLMs can explore in a sample-efficient manner.

 We hope our responses have adequately addressed your concerns. If not, we are happy to engage further in the discussion period.

---

### Decision · Program_Chairs · 2024-09-25

**Decision:**

Accept (poster)

**Comment:**

This paper explores whether large language models (LLMs) can engage in in-context exploration using various prompt designs within multi-armed bandit environments. The study found that only GPT-4, when combined with chain-of-thought reasoning and externally summarized interaction history, demonstrated effective exploration, while other configurations failed.

The reviewers generally acknowledge the paper’s interesting direction and practical value but criticize its lack of deeper insights and innovation in the field. Despite these criticisms, they recognize the potential impact of the paper, which outweighs its weaknesses.